# Sparse actiongen: Accelerating Diffusion Policy with Real-time Pruning

## Abstract

Diffusion Policy has dominated action generation due to its strong capabilities for modeling multi-modal action distributions, but its multi-step denoising processes make it impractical for real-time visuomotor control. Existing caching-based acceleration methods typically rely on *static* schedules that fail to adapt to the *dynamics* of robot-environment interactions, thereby leading to suboptimal performance. In this paper, we propose **S**parse **A**ction**G**en (**SAG**) for extremely sparse action generation. To accommodate the iterative interactions, SAG customizes a rollout-adaptive prune-then-reuse mechanism that first identifies prunable computations globally and then reuses cached activations to substitute them during action diffusion. To capture the rollout dynamics, SAG parameterizes an observation-conditioned diffusion pruner for environment-aware adaptation and instantiates it with a highly parameter- and inference-efficient design for real-time prediction. Furthermore, SAG introduces a one-for-all reusing strategy that reuses activations across both timesteps and blocks in a zig-zag manner, minimizing the global redundancy. Extensive experiments on multiple robotic benchmarks demonstrate that SAG achieves up to $4\times$ generation speedup without sacrificing performance. Project Page: https://sparse-actiongen.github.io/.

## 1 Introduction

Diffusion Policy has gained substantial attention in robotic control, due to its ability to model multi-modal action distributions (Chi et al., 2023; Ze et al., 2024; Liu et al., 2024; 2025c). This capability has led to their widespread adoption in Vision-Language-Action (VLA) models (Black et al., 2024; Shukor et al., 2025; Wen et al., 2025; Liu et al., 2025b; Hou et al., 2025; Li et al., 2025), where they serve as the action generation head for highly dexterous manipulation tasks. However, its massive computational burden in the denoising process makes the action frequency unable to satisfy real-time and smooth robotic control. For example, on a high-end GPU like the RTX 4090, conducting 50 diffusion denoising steps at 1 ms per step for a pick-and-place task takes 50 ms. This restricts the execution frequency to 20 Hz, well below the 50-1000 Hz needed for the Franka robotic arm.

In response to this issue, recent efforts have been made to address the inference bottleneck of Diffusion Policy. For instance, EfficientVLA (Yang et al., 2025) extends the uniform caching schedule to the action diffusion, while BAC (Ji et al., 2025) introduces a task-specific block-wise schedule to improve the caching efficiency. However, these methods typically rely on *static* caching schedules that are predefined offline and remain unchanged throughout the entire rollout process. To examine their limitations, we evaluate multiple fixed caching schedules by applying them to a single different rollout iteration separately in a leave-one-out manner. As shown in Figure 1, *a fixed caching schedule can not consistently accommodate the rollout dynamics*, thereby constraining the achievable tradeoff between performance and efficiency.

To address this, we propose Sparse ActionGen (SAG), a rollout-adaptive acceleration method for extremely sparse action generation. To accommodate the rollout dynamics, SAG discards the fixed-schedule caching procedure and designs an adaptive prune-then-reuse mechanism that operates in concert with the robot-environment interaction. In each iteration, SAG globally identifies prunable computations and reuses cached counterparts to substitute them on the fly throughout generation.

The key to SAG lies in dynamically generating and adjusting the pruning schedule for the changing sparsity patterns of policy inference across rollout iterations, which necessitates a real-time pruner

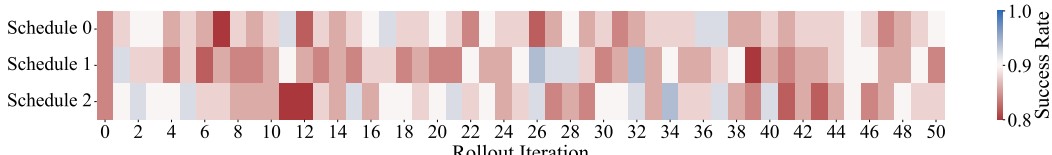

Figure 1: **The performance of fixed schedules on different rollout iterations**. We conduct the experiments on the Square task, in which the robot should push or slide an object so that its center follows a square-shaped path on the table. Observations include: (1) A fixed schedule performs inconsistently across different rollout iterations. (2) The optimal schedules differ among different rollout iterations. Both indicate the limitations of a static caching schedule.

aligned with the interactive visuomotor dynamics. However, existing rule-based approaches (Ji et al., 2025; Liu et al., 2025a) typically require profiling post-forward activations, fundamentally precluding environment-aware adaptation. To resolve this contradiction, SAG parameterizes the pruner to be conditioned on the current observations. In this way, the pruner learns to predict the sparsity pattern a priori before the inference. However, different from previous learning-based works that predefine static computation graphs (Ma et al., 2024a; Zhu et al., 2024), performing pruning in real time inevitably introduces additional cost. To reduce the additional overhead, SAG instantiates the pruner with a parameter- and inference-efficient architectural design that serves all the blocks with a single network and requires only one forward pass for the entire denoising process.

Furthermore, we challenge the widely adopted block-wise caching paradigm (Ji et al., 2025; Ma et al., 2024a; Wimbauer et al., 2024; Yang et al., 2025; Liu et al., 2025a) for its largely overlooked inter-block redundancies. To address this, SAG refines the pipeline from a global allocation perspective. Specifically, for the pruning stage, SAG introduces an end-to-end global sparsity loss that guides the pruner to non-uniformly allocate computational resources across both timesteps and blocks under a strict budget. For the reusing stage, SAG proposes a one-for-all reusing strategy that reuses activations across blocks and timesteps in a zig-zag manner, motivated by the strong similarity observed in the activations across blocks (Figure 4a).

To assess the effectiveness of the proposed method, we evaluate SAG on extensive simulation benchmarks. SAG prunes over 90% computations of Diffusion Policy during the action generation throughout the entire rollout, resulting in $3.6 - 4\times$ speedup on most of the visuomotor tasks without sacrificing the performance. In summary, our main contributions are as follows:

1. We introduce Sparse ActionGen (SAG), a rollout-adaptive acceleration method for extremely fast action generation. SAG addresses the limitations of fixed caching schedules by introducing a real-time prune-then-cache mechanism.

2. SAG proposes a rollout-adaptive diffusion pruner by conditioning it with current observations for the environment-aware adaptation and instantiating it with a highly parameter- and inference-efficient design for real-time prediction.

3. SAG refines the prune–then–reuse pipeline from a global allocation perspective by introducing an end-to-end global sparsity loss to guide computational resource allocation and a one-for-all reusing strategy that promotes cross-block activation reuse.

4. SAG prunes over 90% of computations and delivers up to $4\times$ losslessly acceleration for action generation without updating the model parameters.

## 2  METHODOLOGY

In this section, we introduce SAG, an acceleration method designed to enable extremely fast action generation through rollout-adaptive sparse inference. SAG consists of two key modules: a real-time diffusion pruner that dynamically adjusts pruning schedules based on rollout dynamics, and a one-for-all reusing strategy that reuses computations across denoising steps and blocks. We first analyze the three-level redundancy on the policy model inference in Sec. 2.1. Next, we introduce the real-time diffusion pruner in Sec. 2.2 and describe the one-for-all reusing strategy in Sec. 2.3. Finally, we integrate them into a unified two-stage pipeline.

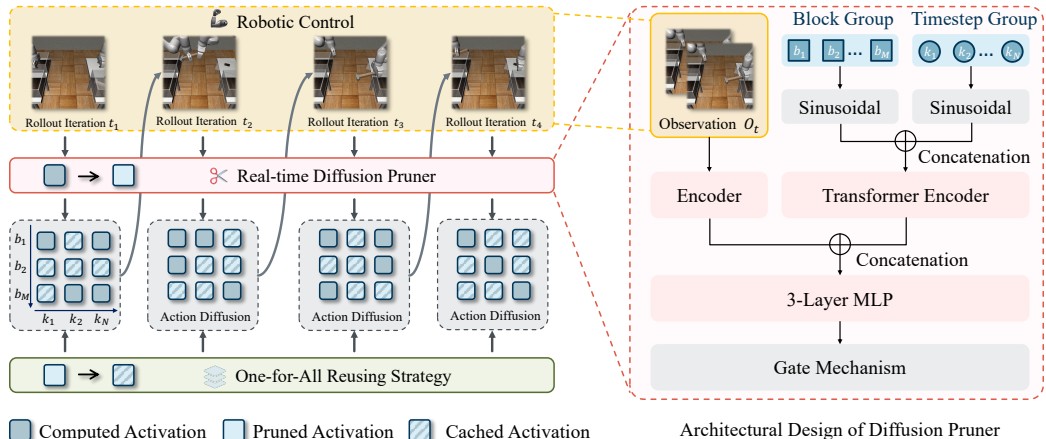

Figure 2: **Framework of Sparse ActionGen.** SAG adopts a prune-then-reuse pipeline coupled with rollout iterations. In each rollout iteration, SAG identifies the prunable computations based on the sparsity pattern predicted by the real-time diffusion pruner. During generation, SAG skips these computations and substitutes them with cached activations in a one-for-all reusing strategy.

## 2.1 PRELIMINARIES

To analyze the redundancy across the action diffusion process, we decompose its dynamics into three nested levels: (1) *rollout-level*, characterizing the iterative interaction between the robot and the environment; (2) *denoising-level*, where actions are generated through multi-step reverse diffusion; and (3) *block-level*, where each denoising step is realized by propagating action tokens through stacked DiT blocks. We formulate the entire process across these levels in turn.

**Rollout-Level: Robot-Environment Interaction.** At rollout iteration $t$, the policy receives the current observation $\mathbf{o}_t$ and produces an action chunk $\mathbf{a}_t$, which is executed by the robot. The environment then evolves under its dynamics and yields the next observation:

$$\mathbf{o}_{t+1} \sim P(\mathbf{o}_{t+1} \mid \mathbf{o}_t, \mathbf{a}_t), \tag{1}$$

where $P(\cdot \mid \mathbf{o}_t, \mathbf{a}_t)$ denotes the environment's state transition probability. This process repeats for multiple iterations until the task is completed.

**Denoising-Level: Diffusion Policy Inference.** In each iteration, the action $\mathbf{a}_t$ is obtained via conditional denoising diffusion (Chi et al., 2023). Starting from Gaussian noise $\mathbf{a}_t^K \sim \mathcal{N}(0, I)$, the policy performs $K$ reverse denoising steps:

$$\mathbf{a}_t^{k-1} = f_\theta\left(\mathbf{a}_t^k, \mathbf{o}_t, k\right), \quad k = K, \dots, 1, \tag{2}$$

where $f_\theta$ is the conditional denoiser. We can therefore represent the entire multi-step denoising process, which maps initial noise to a final action, as the policy $\pi$:

$$\mathbf{a}_t = \pi(\mathbf{o}_t) = \mathbf{a}_t^0 = f_\theta \circ f_\theta \circ \cdots \circ f_\theta(\mathbf{a}_t^K, \mathbf{o}_t). \tag{3}$$

The final $\mathbf{a}_t$ serves as the control action in Eq. 1.

**Block-Level: Diffusion Transformer Forward.** The denoiser $f_\theta$ is parameterized by a DiT. Observations are embedded by a transformer encoder and fused into a transformer decoder with $L$ stacked layers. Each layer $l$ integrates temporal and observation conditioning through three blocks, i.e., self-attention (SA), cross-attention (CA), and a feed-forward network (FFN). Given hidden state $\mathbf{h}_k^{l-1}$ at denoising step $k$, the update is:

$$\mathbf{h}_k^l = \mathbf{h}_k^{l-1} + \text{SA}_k^l + \text{CA}_k^l + \text{FFN}_k^l. \tag{4}$$

The output of layer $l$ is computed by summing the residual outputs of these blocks, and the final representation $\mathbf{h}_k^L$ produces $\mathbf{a}_t^{k-1}$ in Eq. 2.

## 2.2 REAL-TIME DIFFUSION PRUNER

Closed-loop action generation inherently involves multi-round interactions where a diffusion policy progressively generates an entire action trajectory. However, existing caching-based acceleration methods overlook the aforementioned rollout-level dynamics of this process, applying a single, fixed caching schedule. This raises two critical questions: (1) Is the optimal schedule consistent across all rollout iterations? (2) Does a schedule optimized for one specific rollout iteration perform well across the entire trajectory?

To investigate this, we conducted a leave-one-out evaluation on different rollout iterations. Specifically, we randomly generate three fixed caching schedules and apply them to only one rollout iteration at a time to isolate their impact on the final task success rate. As shown in Figure 1, the optimal schedule changes significantly with the rollout progress. Moreover, a schedule that excels at one iteration inevitably degrades performance at others. These findings underscore the limitations of the static schedules and suggest a rollout-adaptive caching mechanism.

The central challenge of such a mechanism is to identify prunable computations based on the sparsity pattern in each rollout iteration. To this end, SAG instantiates the pruner with two essential properties: (1) environment-aware adaptation to predict visuomotor sparsity pattern, and (2) real-time prediction to enable closed-loop visuomotor control. We elaborate on these designs below.

**Environment-aware Adaptation.** Motivated by the preceding analysis, we seek a pruner that dynamically generates optimal pruning schedules that adapt to the rollout dynamics. Ideally, such a schedule should mirror the sparsity pattern of the denoising process in each interaction round. A naive idea would be to approximate this sparsity pattern by profiling activation similarities. However, such rule-based operations rely on full forward passes, negating any potential speedup. To overcome this limitation, we propose a parameterized diffusion pruner $G_\psi$. Instead of profiling post-forward activations for the sparsity pattern, our pruner learns to predict it. In visuomotor tasks, the computational pattern of the action generation process is highly correlated with the visual input snapped from the environment. Therefore, we condition the pruner on the current observation $\mathbf{o}_t$, enabling it to learn a direct mapping from environment inputs to the predicted sparsity pattern. To incorporate sparsity, the pruner predicts a binary pruning mask $\mathcal{M} \in \{0, 1\}^{K \times 3L}$ for all $K$ denoising steps and $3L$ DiT blocks:

$$\mathcal{M} = G_\psi(\mathbf{o}_t), \tag{5}$$

where $\mathcal{M}_{k,b} = 1$ indicates the computation of block $b$ at timestep $k$ is sparse and should be pruned.

**Real-time Prediction.** As the introduced overhead of a parameterized pruner scales with the parameter count and number of its forward passes, SAG minimizes such cost by designing a parameter- and inference-efficient architecture for the pruner $G_\psi$, which generates a global pruning graph in a single forward pass, predicting the sparsity pattern for all $K$ denoising steps and $3L$ blocks in a single forward pass. Specifically, the pruner jointly encodes the group of timestep indices and block indices using sinusoidal positional embeddings (See Figure 3 for quantitative results). Instead of outer summation, these embeddings are concatenated to preserve their informational orthogonality. The resulting spatiotemporal coordinate representation is then processed by a Transformer encoder to capture hierarchical dependencies between temporal and structural signals.

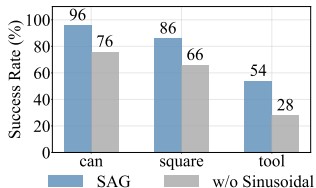

Figure 3: Success rates of SAG w/ and w/o sinusoidal positional encoding.

In parallel, we adopt a visual encoder $\mathrm{Enc}_{obs}$ to process the current observation $\mathbf{o}_t$. The resulting observation embedding is concatenated with the spatiotemporal representation $\mathbf{z}$ to achieve a unified feature vector $\mathbf{h}$:

$$\mathbf{h} = \mathrm{Concat}\left(\mathrm{Enc}_{obs}(\mathbf{o}_t), \mathbf{z}\right). \tag{6}$$

This feature vector, which contains both environmental and structural information, is then projected by a three-layer MLP to produce two-dimensional logits, $\mathbf{l} \in \mathbb{R}^{K \times 3L \times 2}$. The final binary pruning mask $\mathcal{M}$ is obtained by applying an argmax operator along the last dimension of the logits:

$$\mathcal{M} = \mathrm{argmax}\left(\mathrm{MLP}(\mathbf{h})\right). \tag{7}$$

For end-to-end training, we employ the Straight-Through Estimator (STE) (Jang et al., 2016) to enable gradient flow through the non-differentiable argmax operation. Owing to these designs, the proposed pruner operates with exceptional efficiency, incurring a computational overhead of less than 0.3% of the total FLOPs. We present the architecture illustration in Figure 2.

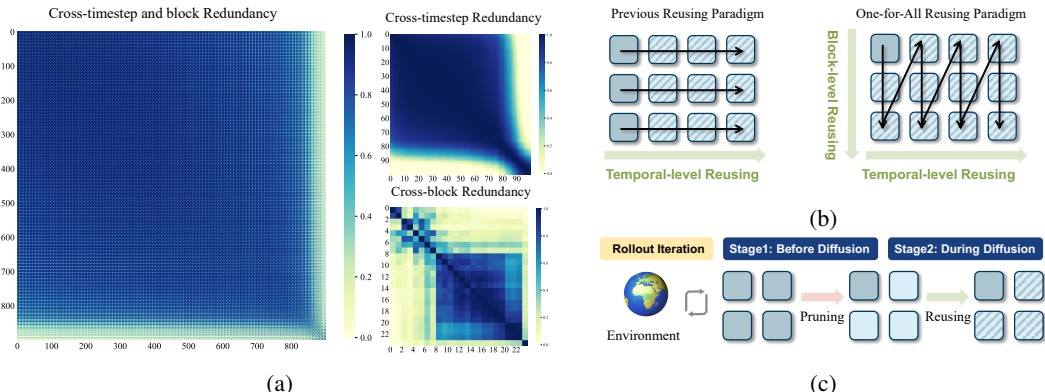

Figure 4: (a) Redundancy patterns at different levels, revealed by calculating the similarities between activations. The left subfigure shows the overall cross-timestep and cross-block redundancy in action generation. The top-right subfigure illustrates the cross-timestep redundancy at the eighth cross-attention block, while the bottom-right subfigure shows the cross-block redundancy at timestep 30. See Appendix A.7 for more details. (b) Comparison of our one-for-all reusing paradigm with the previous block-wise reusing paradigm. The previous approach only reuses activations at the temporal level, whereas our paradigm enables both temporal and block-level reusing. (c) The prune-then-reuse pipeline of SAG, which consists of two stages: (1) *Prune* before the diffusion process to remove redundant computations, and (2) *Reuse* the remaining activations during the diffusion process.

**Optimization Objective.** A key challenge is to guide the learning of the pruner $G_\psi$ such that the pruned policy $\pi'$ preserves task performance while adhering to a computational budget. Our goal is to minimize the deviation between the behaviors of the pruned policy and the ground truth action, regularized by a global sparsity constraint. The overall pruning objective is defined as:

$$\min_\psi \mathbb{E}_{\mathbf{o}_t \sim \mathcal{D}_{\text{ref}}} \left[ \mathcal{L}_{\text{fidelity}}(\pi'(\cdot|\mathbf{o}_t, \mathcal{M}), \mathbf{a}_t) \right] + \beta \mathcal{L}_{\text{sparsity}}(\mathcal{M}). \tag{8}$$

To ensure the pruned policy $\pi'$ aligns with the desired ground-truth behavior, we introduce a policy fidelity loss. Due to the global pruning objective, we directly supervise the pruned policy using a reference dataset $\mathcal{D}_{\text{ref}}$, which is constructed by uniformly sampling approximately 5% of the original training data. This encourages the pruned policy to mimic the expert actions within this representative subset. The loss is formulated as the distance between the policy's output distribution and the ground-truth action $a^*$:

$$\mathcal{L}_{\text{fidelity}} = \mathbb{E}_{(o,a^*) \sim \mathcal{D}_{\text{ref}}} \left[ ||\pi'(\mathbf{o}, \mathcal{M}) - a^*|| \right], \tag{9}$$

where $o$ is a visual state from the reference dataset. To enable non-uniform allocation of computational resources across different timesteps and blocks, we introduce an end-to-end global sparsity loss. This loss directly enforces a target global pruning rate $\rho$, by penalizing the deviation of the overall sparsity degree of the mask $\mathcal{M}$:

$$\mathcal{L}_{\text{sparsity}} = ||\frac{1}{N} \sum_{k=1}^{K} \sum_{b=1}^{B} \mathcal{M}_{k,b} - \rho||. \tag{10}$$

Here, $\mathcal{M}_{k,b} \in [0,1]$ represents the learned gating value for the $b$-th block at $k$-th denoising step, and $N$ is the total number of prunable units, accounting for $K \times 3L$. We employ the Mean Absolute Error (MAE) loss to impose an end-to-end constraint on the model's total computational budget, obviating the need for post-processing algorithms (Zhu et al., 2024) that select computation units based on importance scores. We visualize the real-time pruning rate throughout the rollout in Appendix A.3. By setting $\beta$ to 1, the final optimization objective can be formulated as the sum of two terms:

$$\mathcal{L}_{\text{total}} = \mathcal{L}_{\text{fidelity}} + \mathcal{L}_{\text{sparsity}}. \tag{11}$$

## 2.3 ONE-FOR-ALL REUSING STRATEGY

As for the pruned activations, we prefill them with the activations cached previously. A suitable reusing strategy can maximize the reduction of the caching error, thereby improving the pruning efficiency. Previous caching strategies operate locally within each block, maintaining a caching buffer for each block. However, this strategy overlooks redundancy across blocks. We observe in Figure 4a that activations across different blocks exhibit strong structural similarity, which motivates a one-for-all reusing strategy.

Concretely, instead of maintaining independent cache buffers for each block, we maintain a shared buffer $\tau$ for all the blocks of the same type. This design enables the cross-block reusing in a zig-zag manner (See Fig. 4b), where an activation can be reused for another block in another timestep. The residual computation can be formulated as:

$$h_k^{\lfloor b/3 \rfloor} = h_k^{\lfloor b/3 \rfloor - 1} + (1 - \mathcal{M}_k^b) \cdot d_k^b + \mathcal{M}_k^b \cdot \tau, \tag{12}$$

where $d_k^b$ denotes the computation output of the $b$-th block at $k$-th denosing step. The cache buffer $\tau$ is updated in tandem as:

$$\tau = (1 - \mathcal{M}_k^b) \cdot d_k^b + \mathcal{M}_k^b \cdot \tau. \tag{13}$$

**Pipeline.** We integrate the real-time diffusion pruner and one-for-all reusing strategy into the two-stage pipeline to boost the action diffusion collaboratively. As illustrated in Fig. 4c, at each rollout step, the real-time pruner predicts and decides which computations to prune before the action diffusion, conditioned on the specific observations at each iteration. During the action diffusion, the pruned computations are substituted by the cached activations with the one-for-all reusing strategy, which reuses activations across blocks and timesteps in a zig-zag manner. We adopt the one-for-all reusing strategy on both the training stage and the inference stage of the pruner. The training and inference pipeline is presented as Alg. 1 and 2.

---

**Algorithm 1:** Training

**Input:** Policy $\pi_\theta$, Dataset $\mathcal{D}_{\text{ref}}$, Target Pruning Rate $\rho$, Factor $\beta$
**Output:** Trained Pruner $G_\psi$
Initialize Pruner parameters $\psi$
Freeze Policy parameters $\theta$
**while** *not converged* **do**
  ▷ Generate Pruning Mask
  Sample batch $(\mathbf{o}_t, \mathbf{a}_t^*)$ from $\mathcal{D}_{\text{ref}}$
  $\mathcal{M} = G_\psi(\mathbf{o}_t)$
  ▷ Generate Action
  $\mathbf{a}_{\text{pred}} = \pi_\theta(\mathbf{o}_t, \mathcal{M})$
  $\mathcal{L}_{\text{fidelity}} = ||\pi'(\mathbf{o}_t, \mathcal{M}) - \mathbf{a}_t^*||$   ▷ Eq. 9
  $\mathcal{L}_{\text{sparsity}} = ||\frac{1}{N} \sum_{k,b} \mathcal{M}_{k,b} - \rho||$   ▷ Eq. 10
  $\mathcal{L}_{\text{total}} = \mathcal{L}_{\text{fidelity}} + \beta \mathcal{L}_{\text{sparsity}}$   ▷ Eq. 11
  Update $\psi \leftarrow \psi - \eta \nabla_\psi \mathcal{L}_{\text{total}}$
**end**
**return** $G_\psi$

**Algorithm 2:** Inference

**Input:** Observation $\mathbf{o}_t$, Pruner $G_\psi$, Denoiser $f_\theta$, Steps $K$
**Output:** Action $\mathbf{a}_t$
$\mathcal{M} \leftarrow G_\psi(\mathbf{o}_t)$   ▷ Eq. 5
Initialize global cache $\tau \leftarrow \mathbf{0}$
Sample noise $\mathbf{a}_t^K \sim \mathcal{N}(0, I)$
**for** $k = K$ to $1$ **do**
  $\mathbf{h} \leftarrow \text{Embed}(\mathbf{a}_t^k, k, \mathbf{o}_t)$
  **for** *each block $b$ in $f_\theta$* **do**
    **if** $\mathcal{M}_{k,b} == 1$ **then**
      $\mathbf{h} \leftarrow \mathbf{h} + \tau$   ▷ Eq. 12
    **else**
      $\mathbf{d}_k^b \leftarrow \text{Block}_b(\mathbf{h})$
      $\tau \leftarrow \mathbf{d}_k^b$   ▷ Eq. 13
      $\mathbf{h} \leftarrow \mathbf{h} + \mathbf{d}_k^b$
    **end**
  **end**
  $\mathbf{a}_t^{k-1} \leftarrow \text{Decode}(\mathbf{h})$
**end**
**return** $\mathbf{a}_t^0$

---

## 3 EXPERIMENTS

We first outline the experimental setup, covering models, benchmarks, baselines, and implementation details in Sec. 3.1. Following that, we demonstrate the quantitative results in Sec. 3.2. We then present an ablation study of SAG in Sec. 3.3. Finally, we conduct qualitative experiments on the predicted sparsity pattern in Sec. 3.4.

## 3.1 EXPERIMENTAL SETUP

Table 1: **Benchmark on Proficient Human (PH) demonstration data.** Success rates (%) and speedups are reported. SAG achieves more than a 3.4× speedup across all tasks, with an average performance gain of 13%.

| Method | Success Rate (%, ↑) / Speedup (↑) | | | | |
|---|---|---|---|---|---|
| | Lift | Can | Square | Transport | Tool |
| Full Precision | $100 \pm 0.0$ | $99 \pm 1.2$ | $88 \pm 7.0$ | $78 \pm 3.3$ | $51 \pm 5.9$ |
| DDIM | $100_{\pm0.0}$ (3.32×) | $96_{\pm2.8}$ (3.33×) | $86_{\pm4.9}$ (3.34×) | $76_{\pm5.7}$ (3.30×) | $42_{\pm4.6}$ (3.32×) |
| EfficientVLA | $100_{\pm0.0}$ (3.37×) | $75_{\pm2.1}$ (3.36×) | $86_{\pm3.4}$ (3.32×) | $60_{\pm4.2}$ (3.24×) | $38_{\pm2.7}$ (3.35×) |
| L2C | $100_{\pm0.0}$ (1.26×) | $86_{\pm1.8}$ (1.26×) | $23_{\pm4.1}$ (1.26×) | $66_{\pm3.0}$ (1.33×) | $2_{\pm0.5}$ (1.29×) |
| BAC | $100_{\pm0.0}$ (3.23×) | $94_{\pm1.5}$ (3.42×) | $87_{\pm2.9}$ (3.42×) | $78_{\pm3.6}$ (3.09×) | $\mathbf{51}_{\pm2.8}$ (3.33×) |
| Falcon | $100_{\pm0.0}$ (1.81×) | $85_{\pm2.0}$ (1.13×) | $60_{\pm2.7}$ (1.22×) | $50_{\pm3.3}$ (2.85×) | $42_{\pm1.9}$ (1.17×) |
| SDP | $100_{\pm0.0}$ (1.88×) | $96_{\pm1.4}$ (1.70×) | $85_{\pm2.1}$ (1.70×) | $72_{\pm2.9}$ (1.67×) | $17_{\pm1.2}$ (1.77×) |
| CP | $78_{\pm3.3}$ (15.0×) | $38_{\pm2.8}$ (14.9×) | $22_{\pm4.5}$ (14.8×) | $51_{\pm3.1}$ (10.2×) | $0_{\pm0.0}$ (14.2×) |
| SAG | $\mathbf{100}_{\pm0.0}$ (3.72×) | $\mathbf{98}_{\pm1.6}$ (3.70×) | $\mathbf{89}_{\pm2.5}$ (3.64×) | $\mathbf{85}_{\pm3.3}$ (3.44×) | $50_{\pm2.8}$ (3.65×) |

**Models and Benchmarks.** Following the original settings in Diffusion Policy (Chi et al., 2023), we select the transformer-based Diffusion Policy (DP-T) for our evaluation. The pretrained checkpoints are from source[1]. In simulation settings, we evaluate SAG on DP-T across different robot manipulation tasks with three fixed seeds: Lift, Can, Square, Transport, Tool hang, and Kitchen (Gupta et al., 2020). Demonstration data is sourced from proficient human (PH) and mixed proficient/non-proficient human (MH) teleoperation. More details can be found in Appendix A.2. In real-world settings, we deploy the acceleration methods in a pick-and-release task. The task requires the robot to grasp, pick, and release a cylindrical soft bag.

**Baselines.** We use the DP-T (Chi et al., 2023) model as our Full Precision baseline. For comparison, we benchmark against leading acceleration methods. We reproduce cache-based approaches, including Efficient-VLA (Yang et al., 2025) and BAC (Ji et al., 2025). We further include DDIM (Song et al., 2020), a sampling-efficient noise scheduler, and Falcon (Chen et al., 2025a), which accelerates denoising by reusing intermediate noise from previous actions. Finally, we evaluate the learning-based methods Streaming Diffusion Policy (SDP) (Høeg et al., 2024) and Consistency Policy (CP) (Prasad et al., 2024), which distills the generation into a few denoising steps via consistency distillation, and L2C (Ma et al., 2024a), adapted from image generation to our action-generation setting.

**Implementation Details.** We conduct the simulation experiments on NVIDIA Tesla V100S 32G GPU, equipped with Intel(R) Xeon(R) Silver 4210 CPU. For real-world experiments, we conduct the experiments on Franka Research 3 robot arm with NVIDIA GeForce RTX 4090D 48G. The proposed pruner introduces a hyperparameter pruning rate $\rho$, we empirically set it as 91%. The pruner network is trained for 30 epochs with a batch size of 32 and a learning rate of $1e^{-4}$. We use a training set of 5,600 samples and a validation set of 640 samples for each simulation task and 150 trajectories collected by Gello for the real-world task. We adopt a linear warmup of 4 steps and apply module-specific weight decay, set as $1e^{-4}$ for the observation encoder, $1e^{-3}$ for the transformer encoder, and $1e^{-5}$ for the three-layer MLP. The training process is visualized in Appendix A.4.

## 3.2 QUANTITATIVE RESULTS

To show the advantage of SAG, we compare our method with several SOTA acceleration methods (Yang et al., 2025; Ma et al., 2024a; Ji et al., 2025). We set the cache updating interval $\mathcal{N}$ as 7 for Efficient VLA. For L2C, we adopt the same training configs as SAG for fair comparison. We set the number of cache update steps $\mathcal{S}$ as 10 and the number of selected upstream blocks $k$ as 5 for BAC, following the original paper. We set the inference steps $K$ as 30 for DDIM and 3 for Consistency Policy. For Falcon, we follow the original setup (Chen et al., 2025a).

**Overall Results.** As shown in Tables 1, 2, and 3, SAG demonstrates a superior trade-off between performance and efficiency across all the simulation benchmarks. On the Proficient Human and

---

[1]https://diffusion-policy.cs.columbia.edu/data/experiments/

Table 2: **Benchmark on Mixed Human (MH) demonstration data.** SAG achieves over a $3.7\times$ speedup across all tasks, with an average performance gain of 29%.

| Method | Success Rate (%, ↑) / Speedup (↑) | | | |
|---|---|---|---|---|
| | Lift | Can | Square | Transport |
| Full Precision | $100\pm_{0.0}$ | $93\pm_{6.5}$ | $76\pm_{4.3}$ | $54\pm_{5.2}$ |
| DDIM | $\mathbf{100}\pm_{0.0}$ ($3.31\times$) | $92\pm_{4.9}$ ($3.33\times$) | $78\pm_{2.8}$ ($3.32\times$) | $51\pm_{3.8}$ ($3.30\times$) |
| EfficientVLA | $\mathbf{100}\pm_{0.0}$ ($3.33\times$) | $75\pm_{2.1}$ ($3.34\times$) | $52\pm_{3.0}$ ($3.33\times$) | $0\pm_{0.0}$ ($3.50\times$) |
| L2C | $\mathbf{100}\pm_{0.0}$ ($1.26\times$) | $0\pm_{0.0}$ ($1.26\times$) | $53\pm_{1.9}$ ($1.26\times$) | $46\pm_{2.3}$ ($1.28\times$) |
| BAC | $\mathbf{100}\pm_{0.0}$ ($3.27\times$) | $93\pm_{1.6}$ ($3.43\times$) | $78\pm_{2.8}$ ($3.36\times$) | $29\pm_{3.4}$ ($3.45\times$) |
| Falcon | $\mathbf{100}\pm_{0.0}$ ($1.85\times$) | $84\pm_{2.0}$ ($1.38\times$) | $40\pm_{2.9}$ ($1.54\times$) | $27\pm_{3.1}$ ($2.68\times$) |
| SDP | $\mathbf{100}\pm_{0.0}$ ($1.69\times$) | $\mathbf{94}\pm_{1.4}$ ($1.70\times$) | $77\pm_{2.2}$ ($1.69\times$) | $\mathbf{52}\pm_{3.0}$ ($1.83\times$) |
| CP | $98\pm_{1.0}$ ($15.35\times$) | $15\pm_{2.5}$ ($15.1\times$) | $30\pm_{3.1}$ ($15.4\times$) | $14\pm_{2.0}$ ($11.5\times$) |
| SAG | $\mathbf{100}\pm_{0.0}$ ($3.75\times$) | $\mathbf{94}\pm_{5.7}$ ($3.76\times$) | $\mathbf{79}\pm_{3.4}$ ($3.72\times$) | $50\pm_{1.6}$ ($3.84\times$) |

Table 3: **Benchmark on multi-stage task (Kitchen).** In Kitchen task, $p_x$ is the frequency of interacting with $x$ or more objects, including an openable microwave, four turnable oven burners, an oven light switch, and a freely movable kettle. SAG achieves a lossless $4.03\times$ speedup on this task.

| Method | Success Rate (%, ↑) | | | | Speedup (↑) |
|---|---|---|---|---|---|
| | $\mathrm{Kit}_{p1}$ | $\mathrm{Kit}_{p2}$ | $\mathrm{Kit}_{p3}$ | $\mathrm{Kit}_{p4}$ | |
| Full Precision | $100\pm_{0.0}$ | $100\pm_{0.0}$ | $100\pm_{0.0}$ | $99\pm_{0.6}$ | – |
| DDIM | $100\pm_{0.0}$ | $100\pm_{0.0}$ | $100\pm_{0.0}$ | $99\pm_{0.8}$ | $3.37\times$ |
| EfficientVLA | $20\pm_{2.3}$ | $2\pm_{0.8}$ | $0\pm_{0.0}$ | $0\pm_{0.0}$ | $3.71\times$ |
| L2C | $\mathbf{100}\pm_{0.0}$ | $\mathbf{100}\pm_{0.0}$ | $\mathbf{100}\pm_{0.0}$ | $97\pm_{1.2}$ | $1.28\times$ |
| BAC | $\mathbf{100}\pm_{0.0}$ | $\mathbf{100}\pm_{0.0}$ | $97\pm_{1.6}$ | $90\pm_{2.9}$ | $3.66\times$ |
| Falcon | $\mathbf{100}\pm_{0.0}$ | $\mathbf{100}\pm_{0.0}$ | $99\pm_{0.6}$ | $\mathbf{99}\pm_{0.6}$ | $3.01\times$ |
| SDP | $\mathbf{100}\pm_{0.0}$ | $\mathbf{100}\pm_{0.0}$ | $\mathbf{100}\pm_{0.0}$ | $\mathbf{99}\pm_{0.6}$ | $1.63\times$ |
| CP | $76\pm_{2.5}$ | $63\pm_{4.3}$ | $46\pm_{4.5}$ | $12\pm_{6.8}$ | $31.4\times$ |
| SAG | $\mathbf{100}\pm_{0.0}$ | $\mathbf{100}\pm_{0.0}$ | $\mathbf{100}\pm_{0.0}$ | $\mathbf{99}\pm_{0.6}$ | $4.03\times$ |

Mixed Human datasets, SAG improves the average success rates to 84% and 81%, respectively. We also observe that SAG achieves substantial speedups of $3.63\times$ and $3.77\times$ with an extreme pruning rate exceeding 90%. The advantage is most pronounced on the multi-stage Kitchen task, where SAG achieves 100% success rate while delivering the highest speedup of $4.03\times$. The latency is detailed in Appendix A.5. As shown in Fig. 5, SAG also demonstrates robust performance in real-world settings, which impose strict requirements for real-time and high-fidelity action generation. SAG achieves the best success rate of 54% and the second-best inference frequency of 45.1%. More direct comparisons can be found on the project website.

**Comparison with SOTA baselines.** Notably, all the baselines fail to achieve lossless performance with comparable acceleration gains. EfficientVLA uses a fixed caching schedule for any input, particularly on the Kitchen task, where its success rate falls to 3%. L2C's rigid policy of caching every two steps inherently caps its acceleration, resulting in a modest $1.28\times$ speedup. BAC cannot adapt to evolving rollout dynamics and remains tied to a block-wise caching paradigm, performing poorly on complex tasks such as transport. The excessive compression rate seriously impairs the performance of CP, while Falcon and SDP achieve sub-optimal speedup due to coarse-grained pruning strategies. In contrast, SAG achieves environment-aware, real-time pruning to adapt to the rollout dynamics and achieves a global prune-then-reuse pipeline to preserve fidelity of generated actions under extremely high sparsity.

### 3.3 ABLATION STUDY

To verify the effectiveness of the key design choices of SAG, we conduct a series of ablation experiments, with results summarized in Table 4.

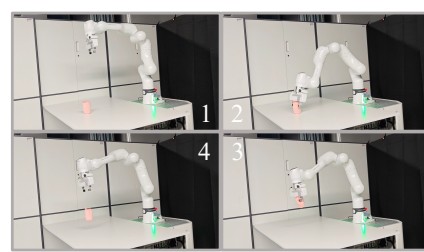 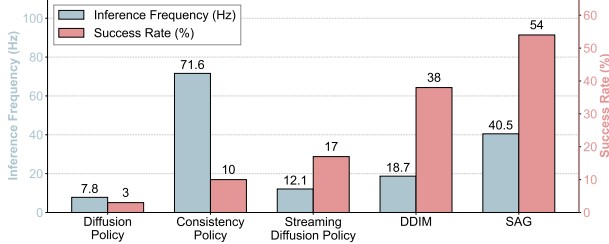

Figure 5: Left: Pick-and-Release Task Setup. Right: Real-world evaluation results, measured by inference frequency and success rate.

Table 4: **Ablation Study.** We evaluate three key design choices of SAG: the real-time diffusion pruner, the one-for-all reusing strategy, and the global sparsity loss. Removing any of them results in a significant drop in success rate, confirming their critical role in maintaining efficiency and accuracy.

| Method | Success Rate (%, ↑) | | | | | Average |
|---|---|---|---|---|---|---|
| | $\text{Lift}_{ph}$ | $\text{Can}_{ph}$ | $\text{Square}_{ph}$ | $\text{Transport}_{ph}$ | Tool | |
| w/o Real-time Diffusion Pruner | **100** | 92 | 90 | 80 | 48 | 82 |
| w/o One-for-All Reusing Strategy | **100** | **94** | **94** | 0 | 36 | 65 |
| w/o Global Sparsity Loss | **100** | 86 | 0 | 16 | 40 | 48 |
| SAG | **100** | **94** | **94** | **84** | **54** | **85** |

| Method | Success Rate (%, ↑) | | | | | Average |
|---|---|---|---|---|---|---|
| | $\text{Lift}_{mh}$ | $\text{Can}_{mh}$ | $\text{Square}_{mh}$ | $\text{Transport}_{mh}$ | Kitchen | |
| w/o Real-time Diffusion Pruner | **100** | 94 | 76 | 46 | 99 | 83 |
| w/o One-for-All Reusing Strategy | **100** | 94 | 80 | 48 | 97 | 84 |
| w/o Global Sparsity Loss | **100** | 90 | 0 | 0 | 93 | 57 |
| SAG | **100** | **96** | **86** | **62** | **100** | **89** |

**Effectiveness of the Real-time Diffusion Pruner.** For comparison, we adopt the diffusion pruner to generate a fixed caching schedule and substitute the pruner with this schedule throughout the entire rollout. The results show that the substitution leads to a noticeable drop in performance, with the average success rate decreasing from 89% to 83% on the MH and Kitchen tasks. This degradation highlights that achieving environmental awareness and real-time pruning is the key to realizing efficient and lossless action generation.

**Effectiveness of the One-for-All Reusing Strategy.** We replace the one-for-all reusing strategy with the widely adopted block-wise reusing strategy to isolate its impact on the overall performance. The results show that the one-for-all reusing strategy improves the performance significantly by 12% on average. This demonstrates that the proposed strategy reduces both the cross-block and cross-timestep redundancy.

**Effectiveness of the Global Sparsity Loss.** We assess the global sparsity loss by substituting it with a block-wise loss, which assigns a uniform computational budget to every block. The removal of global sparsity loss leads to a catastrophic collapse in performance across all tasks, with the average success rate dropping by 37% on PH tasks and 32% on MH tasks. Notably, performance on several tasks, such as Square and $\text{Transport}_{mh}$, drops to 0%. This outcome underscores that allocating as many computational resources as possible to important blocks is critical to achieving extreme sparsity.

### 3.4 QUALITATIVE RESULTS.

We visualize the sparsity patterns predicted by the pruner during the robot moving a kettle in Figure 6. We observe that most computations are highly sparse, as shown by the large white regions. Moreover,

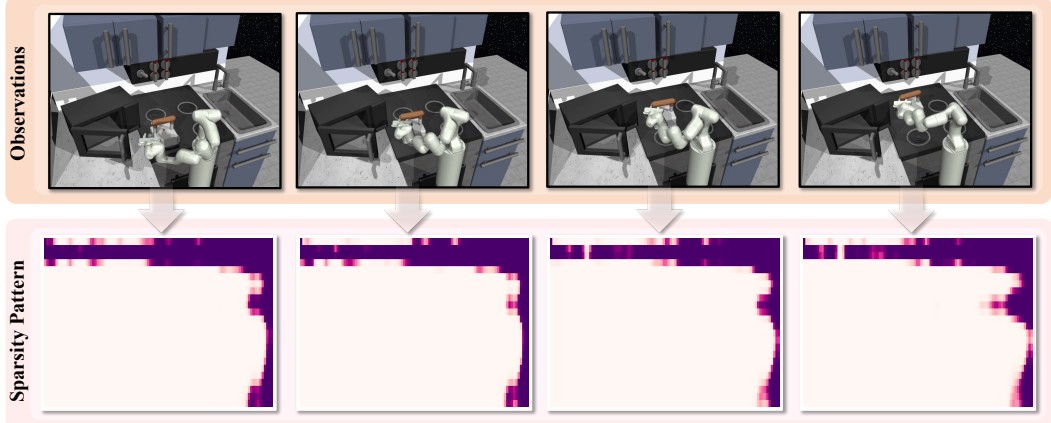

Figure 6: **Predicted sparsity patterns across rollout iterations when the robot moves the kettle**. A coordinate point (x,y) in the sparsity pattern figures represents the computation of the y-th block at the x-th timestep. Lighter colors indicate higher computation sparsity.

the pattern shifts with each observation, demonstrating the pruner's ability to adapt to the evolving rollout dynamics. Interestingly, the computations become dense in the top and right areas, indicating that the upper blocks and later denoising steps exhibit more significant changes in activation patterns. We also present details of sparsity patterns for different tasks in Appendix A.6.

## 4   RELATED WORK

**Diffusion Models Caching.**   Caching has emerged as an effective way to accelerate diffusion inference by reusing redundant activations. Early methods such as DeepCache (Ma et al., 2024b) focus on U-Net backbones, while more recent approaches (Selvaraju et al., 2024; Ma et al., 2024a; Chen et al., 2024; Zou et al., 2025) extend caching to transformer-based diffusion models. These methods typically operate at a coarse granularity, with all the blocks sharing a uniform caching schedule (Selvaraju et al., 2024; Ma et al., 2024b), i.e., updating the cache at uniform intervals. Despite some works extending this schedule in a finer architectural granularity, they either require extra training (Ma et al., 2024a) or are specifically designed for the patterns of the image generation process (Chen et al., 2024).

**Diffusion Policy Acceleration.**   DiT-based diffusion policies use a large multimodal transformer as the denoiser and directly denoise continuous action chunks conditioned on visual observations. One-Step Diffusion Policy (Wang et al., 2024) converts a pretrained policy into a single-step action generator. Consistency Policy (Prasad et al., 2024) and SDM Policy (Jia et al., 2024) distill a Diffusion Policy into a faster student policy by enforcing self-consistency along diffusion trajectories. VDD (Zhou et al., 2024) compresses pretrained diffusion policies into Mixture-of-Experts models using a variational objective. In terms of caching-based methods, Streaming Policy Høeg et al. (2024), RNR-DP Chen et al. (2025b), and Falcon (Chen et al., 2025a) reuse partially denoised trajectories in the previous iterations, while EfficientVLA (Yang et al., 2025) adopts a uniform caching schedule, i.e., updating the cached features at fixed intervals. Block-wise Adaptive Caching (Ji et al., 2025) accelerates policy inference by caching intermediate action features at a fine-grained block level.

## 5   CONCLUSION

In this work, we propose Sparse ActionGen (SAG) for sparse action diffusion. SAG boosts action generation with two key designs. First, SAG designs a real-time pruner that adapts caching schedules to the rollout dynamics during robot–environment interaction. Second, SAG develops a global prune-then-reuse pipeline that enables both global allocation of computational resources and global reuse of activations across blocks and timesteps. Extensive experiments on diverse benchmarks demonstrate the superior efficiency and effectiveness of our approach.

## 6 ETHICS STATEMENT

In this work, all experiments are conducted on publicly available robotic simulation datasets, including RoboMimic, and Franka Kitchen. The demonstration data were originally collected under published protocols via teleoperation or scripted policies, and no new human subjects or personal data were involved. Our study does not raise concerns regarding privacy, discrimination, or bias, since the tasks are synthetic robotic manipulation environments. The proposed method, Sparse ActionGen, focuses solely on improving computational efficiency of diffusion-based robotic policies, and we identify no risks of rights infringement, legal non-compliance, or unethical use.

## 7 REPRODUCIBILITY STATEMENT

We have made extensive efforts to ensure reproducibility of our results. The datasets used in this study are publicly available (RoboMimic, Push-T, Block Pushing, Franka Kitchen), and detailed descriptions of benchmarks, data sources, and task settings are provided in Appendix A.2. The proposed methodology, including the real-time diffusion pruner and one-for-all reusing strategy, is described in Section 2 with training and implementation details reported in Section 3.1. Additional ablation studies, pruning rates, and loss curves are included in the Appendix A.3, A.7, A.8, A.6. All technical details necessary for replication, including hyperparameters, architectures, and evaluation protocols, are provided. The full implementation and instructions will be released on an open-source GitHub repository to facilitate replication of all experiments.

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

# A  APPENDIX

## A.1  THE USE OF LARGE LANGUAGE MODELS (LLMS)

In this work, Large Language Models (LLMs) were used solely to polish the language for clarity and readability. No LLMs were employed for idea generation, experimental design, data analysis, or any other part of the research process.

## A.2  MORE DETAILS ON THE BENCHMARK

### A.2.1  DATASETS

**Franka Kitchen.** This dataset originates from the Relay Policy Learning framework proposed by Gupta et al. (Gupta et al., 2019), featuring 566 VR tele-operated demonstrations of multi-step manipulation tasks in a simulated kitchen using a 9-DoF Franka Panda arm. The goal is to execute as many demonstrated tasks as possible, regardless of order, showcasing both short-horizon and long-horizon multimodality (Chi et al., 2023).

**RoboMimic.** This dataset, introduced by Mandlekar et al. (2021), covers five manipulation tasks. Each task includes a Proficient-Human (PH) teleoperated demonstration set, and four of the tasks additionally offer Mixed-Human (MH) sets combining proficient and non-proficient operators (9 variants in total). The PH data were recorded by a single operator via the RoboTurk platform, whereas the MH sets were collected from six different operators using the same system.

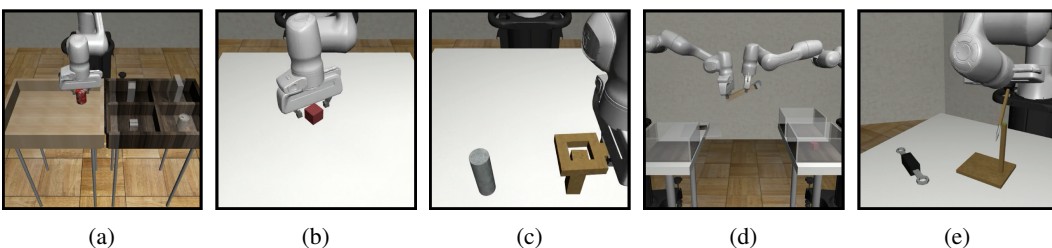

|        |        |        |        |        |
|--------|--------|--------|--------|--------|
| (a)    | (b)    | (c)    | (d)    | (e)    |

Figure 7: Visualizations of different tasks. (a) Can (b) Lift (c) Square (d) Transport (e) Tool_hang.

Fig. 7 illustrates the five subtasks in the RoboMimic Image dataset. Below we describe each subtask.

- **Can** (Fig. 7a): The robot must grasp a cylinder-shaped object and placing it into a bin. This subtask tests precise grasp planning and fingertip control under varying object poses (Mandlekar et al., 2021).

- **Lift** (Fig. 7b): The manipulator picks up a heavier, irregularly shaped object (e.g. a small box) and raises it to a designated height. It evaluates the policy's ability to modulate grip force and maintain stable trajectories (Mandlekar et al., 2021).

- **Square** (Fig. 7c): The agent must push or slide an object so that its center follows a square-shaped path on the table. This challenges both straight-line control and precise cornering maneuvers (Mandlekar et al., 2021).

- **Transport** (Fig. 7d): The agent must learn bimanual maneuvers to transfer a hammer from a closed container on a shelf to the target bin on another shelf. It tests coordinated lifting and translational motion under variable loads (Mandlekar et al., 2021).

- **Tool_Hang** (Fig. 7e): The robot arm must learn high-precision manipulation behaviors to assemble a frame by inserting a hook into a narrow base. This requires fine-tuned wrist orientation and insertion accuracy (Mandlekar et al., 2021).

### A.2.2 PRE-TRAINED CHECKPOINTS

We use the pre-trained checkpoints of `diffusion_policy_transformer` model provided by Diffusion Policy (Chi et al., 2023), where checkpoints of image-based tasks are stored under link[2] and those of multi-stage tasks are stored under link[3].

## A.3 PRUNING RATE

To further verify the stability of SAG's acceleration effect, we conduct an evaluation of its real-time pruning behavior during inference under a target pruning rate of 91%. The experimental results demonstrate that SAG maintains a highly stable real-time pruning process, with the observed real-time pruning rate deviating by no more than 1.6% from the target.

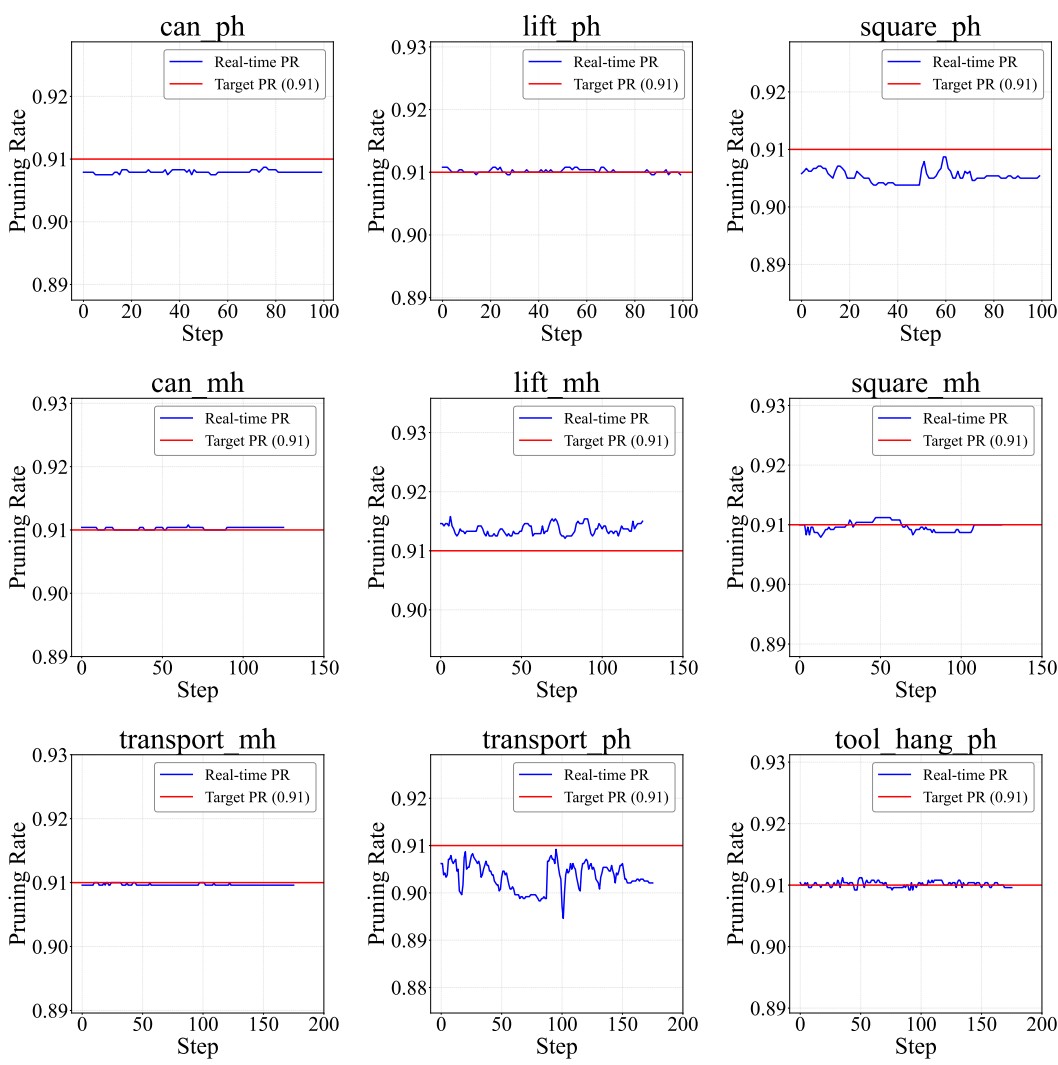

Figure 8: Real-time pruning rate throughout the entire rollout.

## A.4 LOSS CURVES

We present training curves of SAG under target pruning ratios of 80 %, 85 %, and 91 %. As shown in Fig. 9 and Fig. 10, the training loss decreases smoothly and converges to low values, while the

---

[2]https://diffusion-policy.cs.columbia.edu/data/experiments/image/
[3]https://diffusion-policy.cs.columbia.edu/data/experiments/low_dim/

validation loss remains stable with only minor fluctuations. These results provide evidence that SAG can achieve significant inference acceleration without incurring measurable degradation in predictive performance, thereby exhibiting robustness and reliability across a broad spectrum of pruning intensities.

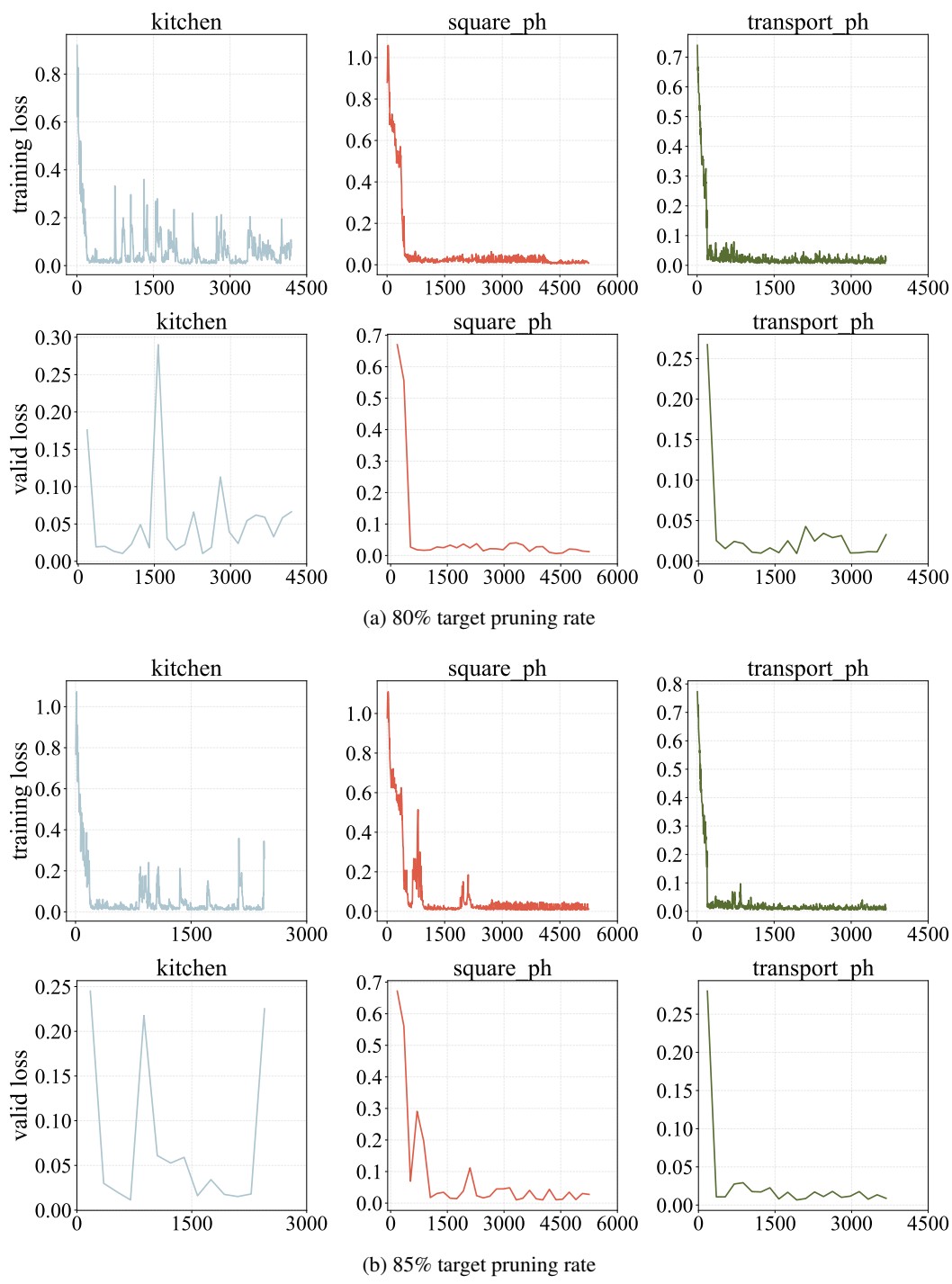

Figure 9: Loss curves under different target pruning rates (part 1).

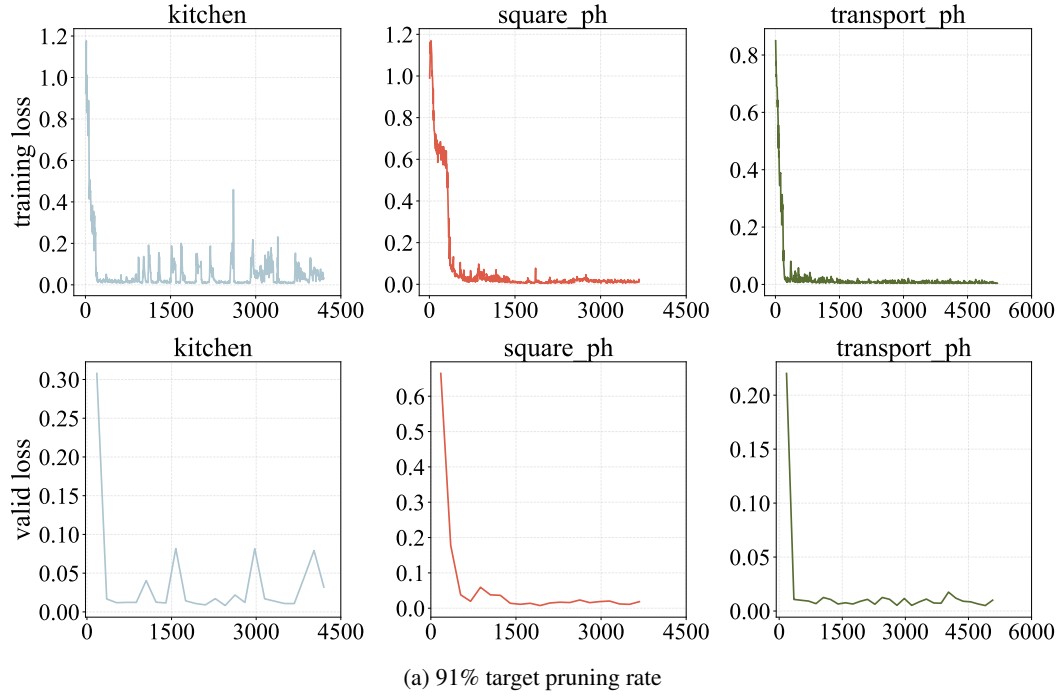

(a) 91% target pruning rate

Figure 10: Loss curves under different target pruning rates (part 2).

## A.5 DETAILED LATENCY ON SIMULATION EXPERIMENTS

We present the detailed latency of SAG on the simulation benchmarks in Table 5. The speedup is calculated by dividing the ratio of the wall-clock time after acceleration by the original time.

Table 5: **Inference Latency.**

| Method | Latency (s, ↓) | | | | | Average |
|---|---|---|---|---|---|---|
| | Lift | Can | Square | Transport | Tool | |
| Full Precision | 1.55 | 1.54 | 1.56 | 1.61 | 1.56 | 1.66 |
| DDIM | 0.47 | 0.46 | 0.47 | 0.49 | 0.47 | 0.47 |
| EfficientVLA | 0.46 | 0.46 | 0.47 | 0.50 | 0.47 | 0.50 |
| L2C | 1.23 | 1.22 | 1.24 | 1.21 | 1.21 | 1.29 |
| BAC | 0.48 | 0.45 | 0.46 | 0.52 | 0.47 | 0.50 |
| Falcon | 0.86 | 1.36 | 1.28 | 0.57 | 1.33 | 1.16 |
| SDP | 0.82 | 0.91 | 1.92 | 0.96 | 0.88 | 0.95 |
| CP | 0.10 | 0.10 | 0.11 | 0.16 | 0.11 | 0.12 |
| **SAG** | 0.42 | 0.42 | 0.43 | 0.47 | 0.43 | 0.46 |

## A.6 MORE DETAILS ON SPARSITY PATTERN VISUALIZATION

In this section, we present the predicted sparsity patterns by the pruner across different tasks. As shown in Fig. 11, the patterns differ for different tasks, showcasing that SAG can capture task-specific sparsity patterns, thereby achieving environment-aware adaptivity.

## A.7 MORE DETAILS ON REDUNDANCY VISUALIZATION

To investigate the block-level redundancy within the DiT module of the diffusion policy, we conduct a comprehensive cosine-similarity analysis across every denoising step. In particular, for each

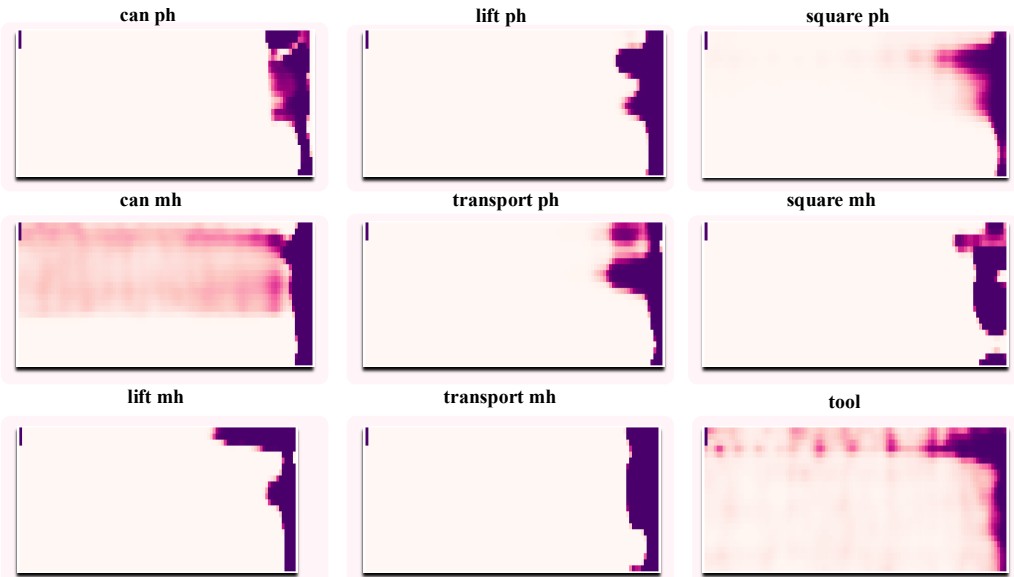

Figure 11: Predicted sparsity patterns for different tasks.

individual denoising step, we examine all transformer layers of the DiT module—specifically the eight layers of self-attention, cross-attention, and feed-forward (FFN) blocks.

The visualization results reveal that every step consistently contains a substantial proportion of redundant computations within these components, and that the pairwise similarity patterns remain highly consistent across the entire denoising process.

Such pronounced cross-step and intra-layer correlations indicate that many block operations contribute little new information, suggesting that the effective representational capacity of the module is considerably lower than its nominal parameter count implies.

## A.8    MORE DETAILS ON SINUSOIDAL POSITIONAL ENCODING

Since step and block identifiers serve as the primary inputs to SAG, their representation plays a critical role in determining pruning quality. To examine this effect, we conduct experiments under a target pruning rate of 91% comparing sinusoidal positional encoding with a straightforward learned embedding (nn.Embedding) scheme. The results clearly show that sinusoidal positional encoding delivers substantially superior performance, highlighting the importance of well-structured positional information for stable and accurate pruning.

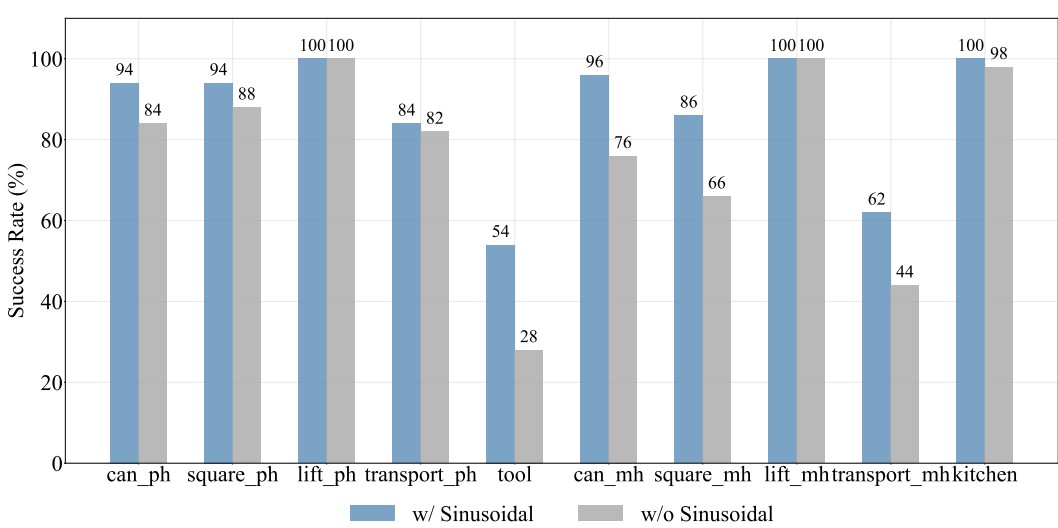

(a) steps 10, 20, and 30

(b) steps 40, 50, and 60

(c) steps 70, 80, and 90

Figure 12: Redundancy across steps 10–90.

Figure 13: Task success rate (%) across multiple tasks for different positional encodings.

