# OpenReview forum: "Sparse ActionGen: Accelerating Diffusion Policy with Real-time Pruning"
_ICLR.cc/2026/Conference — Submitted to ICLR 2026_

### Official Review · Reviewer_ZwXH · 2025-10-22

**Soundness:** 3
**Presentation:** 3
**Contribution:** 3
**Rating:** 2
**Confidence:** 4

**Summary:**

This paper introduces Sparse ActionGen (SAG), a novel method to accelerate diffusion policies for robotic control. The authors argue that existing acceleration techniques based on static caching schedules are suboptimal because they cannot adapt to the changing dynamics of robot-environment interactions during a task rollout. To address this, SAG proposes a rollout-adaptive, prune-then-reuse mechanism. The core contributions include (1) a real-time, observation-conditioned diffusion pruner that dynamically predicts which computations can be skipped in each rollout iteration; (2) a "one-for-all" reusing strategy that shares a cache across both denoising timesteps and model blocks to maximize activation reuse; and (3) a global sparsity loss to train the pruner, enabling non-uniform resource allocation under a predefined computational budget. The authors evaluate SAG on several visuomotor manipulation tasks, reporting up to a 4x speedup over a full-precision baseline while claiming to maintain or even improve task performance.

**Strengths:**

- The paper is well-written, clearly structured, and easy to follow. The problem is framed effectively by decomposing the process into rollout, denoising, and block levels. The figures, particularly the framework diagram (Figure 2), are helpful in understanding the overall approach.
- The experimental evaluation is extensive, covering multiple benchmarks (RoboMimic, Kitchen). The ablation studies are thorough and effectively validate the importance of each proposed component.

**Weaknesses:**

- **Incomplete Baseline Comparison**: The paper's experimental validation suffers from a fundamental flaw. SAG is a training-based acceleration method, requiring an explicit training phase for the pruner. However, the authors almost exclusively compare it against caching-based methods (EfficientVLA, BAC) which are largely training-free. This is an inappropriate and misleading comparison. The most relevant and powerful baselines are other training-based acceleration techniques, such as policy distillation methods. The absence of a comparison to Consistency Policy (RSS 2024) [1] is a fatal omission. Without demonstrating superiority over such methods, the central motivation for proposing this complex pruning mechanism is fundamentally questionable. The authors must justify why their approach is necessary when simpler and potentially more effective distillation-based solutions exist.

- **Inadequate Literature Review**: The literature review in the "Related Work" section is incomplete. The authors fail to cite several highly relevant and recent works on diffusion policy acceleration, such as Streaming Diffusion Policy (ICRA 2025) [2] and Falcon (ICML 2025) [3]

- **Questionable Empirical Claims and Analysis**: The paper reports that SAG consistently outperforms the full-precision baseline (e.g., 94% vs. 90% in Table 1 (Square), 84% vs. 80% in Table 1 (Transport), and 54% vs. 50% in Table 1 (Tool). This is a counterintuitive result that is presented without any plausible explanation. Such a claim requires extraordinary evidence, yet the paper provides no statistical analysis (e.g., standard deviations over multiple seeds) to verify that this improvement is not simply an artifact of evaluation variance. This unsubstantiated claim undermines the credibility of the results.

- **Lack of Real-World Validation**: As a robotics paper, the paper does not have any real-world experiments.


[1] Consistency Policy: Accelerated Visuomotor Policies via Consistency Distillation: https://arxiv.org/abs/2405.07503

[2] Fast Policy Synthesis with Variable Noise Diffusion Models: https://arxiv.org/abs/2406.04806

[3] Falcon: Fast Visuomotor Policies via Partial Denoising: https://arxiv.org/abs/2503.00339

**Questions:**

1. Your method is a training-based approach. Why did you not compare it against other SOTA training-based acceleration methods, most notably Consistency Policy? Please provide a detailed comparison. What are the specific advantages (e.g., training cost, final performance, inference speed) of your prune-then-reuse paradigm over policy distillation?

2. Given that this paper is positioned to address a challenge in robotics, can you provide the results in the real-world experiments?

3. The results in Tables 1 and 2 show that SAG achieves a higher success rate than the full-precision model, which is paradoxical for a method based on pruning. Could you offer a concrete hypothesis for this phenomenon? More importantly, please provide standard deviations across multiple random seeds for both your method and the baselines to demonstrate that this performance gain is statistically significant.

4. Can you provide some demos on the comparison of your method and diffusion policy?

I will adjust my future rating based on the improvements made. If I have misunderstood any points, I am open to discussion.

---

> ### Author Response · Authors · 2025-12-04
>
> We thank the reviewer for the feedback. In this revision, we have significantly expanded our experimental suite to include the suggested training-based methods and real-world validations.
>
> ---
>
> **W1 & Q1. Incomplete baseline comparison**
>
> **A1.** We have included all the suggested baselines i.e., Consistency Policy, Streaming Policy, and Falcon to our experiments. The results detailed in Tables 1, 2, and 3 below reveal a critical trade-off that validates the necessity of SAG:
>
> While distillation-based methods like CP achieve extreme speedups that up to 15$\times$, they suffer from severe performance degradation on complex, multi-modal tasks. For instance, CP achieves only 14% success on Transport (vs. SAG’s 50%) and 12% on Kitchen (vs. SAG’s 99%). This suggests that distillation often leads to mode collapse in high-precision, long-horizon scenarios.
>
> Falcon and SDP attempt to accelerate inference via feature reuse. However, their coarse-grained reuse strategies result in sub-optimal performance compared to SAG's fine-grained, prune-then-reuse mechanism (e.g., SAG outperforms Falcon by +35% on Transport).
>
> SAG occupies a unique "sweet spot," delivering a $3.6\times$ – $4.0\times$ speedup while maintaining lossless performance comparable to the Full Precision baseline. This demonstrates that for high-fidelity control, preserving the original diffusion backbone via sparse inference is superior to distilling it away.
>
> **Table1: PH Benchmark**
>
> | Method | Lift | Can | Square | Transport | Tool |
> | :--- | :--- | :--- | :--- | :--- | :--- |
> | Full Precision | 100$\pm$0.0 | 99$\pm$1.2 | 88$\pm$7.0 | 78$\pm$3.3 | 51$\pm$5.9 |
> | DDIM | **100$\pm$0.0** (3.32$\times$) | 96$\pm$2.8 (3.33$\times$) | 86$\pm$4.9 (3.34$\times$) | 76$\pm$5.7 (3.30$\times$) | 42$\pm$4.6 (3.32$\times$) |
> | EfficientVLA | **100$\pm$0.0** (3.37$\times$) | 75$\pm$2.1 (3.36$\times$) | 86$\pm$3.4 (3.32$\times$) | 60$\pm$4.2 (3.24$\times$) | 38$\pm$2.7 (3.35$\times$) |
> | L2C | **100$\pm$0.0** (1.26$\times$) | 86$\pm$1.8 (1.26$\times$) | 23$\pm$4.1 (1.26$\times$) | 66$\pm$3.0 (1.33$\times$) | 2$\pm$0.5 (1.29$\times$) |
> | BAC | **100$\pm$0.0** (3.23$\times$) | 94$\pm$1.5 (3.42$\times$) | 87$\pm$2.9 (3.42$\times$) | 78$\pm$3.0 (3.09$\times$) | 51$\pm$2.8 (3.33$\times$) |
> | Falcon | **100$\pm$0.0** (1.81$\times$) | 85$\pm$2.0 (1.13$\times$) | 60$\pm$2.7 (1.22$\times$) | 50$\pm$3.3 (2.85$\times$) | 42$\pm$1.9 (1.17$\times$) |
> | SDP | **100$\pm$0.0** (1.88$\times$) | 96$\pm$1.4 (1.70$\times$) | 85$\pm$2.1 (1.70$\times$) | 72$\pm$2.9 (1.67$\times$) | 17$\pm$1.2 (1.77$\times$) |
> | CP | 78$\pm$3.3 (15.0$\times$) | 38$\pm$2.8 (14.9$\times$) | 22$\pm$4.5 (14.8$\times$) | 51$\pm$3.1 (10.2$\times$) | 0$\pm$0.0 (14.2$\times$) |
> | **SAG** | **100$\pm$0.0** (3.72$\times$) | **98$\pm$1.6** (3.70$\times$) | **89$\pm$2.5** (3.64$\times$) | **85$\pm$3.3** (3.44$\times$) | **50$\pm$2.8** (3.65$\times$) |
>
> **Table2: MH Benchmark**
>
> | Method | Lift | Can | Square | Transport |
> | :--- | :--- | :--- | :--- | :--- |
> | Full Precision | 100$\pm$0.0 | 93$\pm$6.5 | 76$\pm$4.3 | 54$\pm$5.2 |
> | DDIM | **100$\pm$0.0** (3.31$\times$) | 92$\pm$4.9 (3.33$\times$) | 78$\pm$2.8 (3.32$\times$) | 51$\pm$3.8 (3.30$\times$) |
> | EfficientVLA | **100$\pm$0.0** (3.33$\times$) | 75$\pm$2.1 (3.34$\times$) | 52$\pm$3.0 (3.33$\times$) | 0$\pm$0.0 (3.50$\times$) |
> | L2C | **100$\pm$0.0** (1.26$\times$) | 53$\pm$1.9 (1.26$\times$) | 53$\pm$1.9 (1.26$\times$) | 46$\pm$2.3 (1.28$\times$) |
> | BAC | **100$\pm$0.0** (3.27$\times$) | 93$\pm$3.4 (3.36$\times$) | 87$\pm$2.8 (3.36$\times$) | 29$\pm$3.3 (3.45$\times$) |
> | Falcon | **100$\pm$0.0** (1.85$\times$) | 84$\pm$2.0 (1.38$\times$) | 72$\pm$2.9 (1.54$\times$) | 27$\pm$3.1 (2.68$\times$) |
> | SDP | **100$\pm$0.0** (1.69$\times$) | 94$\pm$1.1 (1.70$\times$) | 77$\pm$2.2 (1.69$\times$) | 52$\pm$3.0 (1.83$\times$) |
> | CP | 98$\pm$1.0 (15.3$\times$) | 30$\pm$3.1 (15.4$\times$) | 30$\pm$3.1 (15.4$\times$) | 14$\pm$2.0 (11.5$\times$) |
> | **SAG** | **100$\pm$0.0** (3.75$\times$) | **94$\pm$5.7** (3.76$\times$) | **79$\pm$3.4** (3.72$\times$) | **50$\pm$1.6** (3.84$\times$) |
>
>
> **Table3: Kitchen Task**
>
> | Method | Kit_p1 | Kit_p2 | Kit_p3 | Kit_p4 | Speedup |
> | :--- | :--- | :--- | :--- | :--- | :--- |
> | Full Precision | 100$\pm$0.0 | 100$\pm$0.0 | 100$\pm$0.0 | 99$\pm$0.6 | – |
> | DDIM | 100$\pm$0.0 | 100$\pm$0.8 | 100$\pm$0.0 | 99$\pm$0.8 | 3.37$\times$ |
> | EfficientVLA | 20$\pm$2.3 | 2$\pm$0.8 | 0$\pm$0.0 | 0$\pm$0.0 | 3.71$\times$ |
> | L2C | **100$\pm$0.0** | **100$\pm$0.0** | **100$\pm$0.0** | 97$\pm$1.2 | 1.28$\times$ |
> | BAC | **100$\pm$0.0** | 97$\pm$1.6 | 97$\pm$1.6 | 97$\pm$1.2 | 3.66$\times$ |
> | Falcon | **100$\pm$0.0** | **100$\pm$0.0** | 99$\pm$0.6 | 99$\pm$0.6 | 3.01$\times$ |
> | SDP | **100$\pm$0.0** | **100$\pm$0.0** | **100$\pm$0.0** | 99$\pm$0.6 | 1.63$\times$ |
> | CP | 76$\pm$2.5 | 63$\pm$4.3 | 46$\pm$4.5 | 12$\pm$6.8 | 31.4$\times$ |
> | **SAG** | **100$\pm$0.0** | **100$\pm$0.0** | **100$\pm$0.0** | **99$\pm$0.6** | **4.03$\times$** |

---

> ### Author Response · Authors · 2025-12-04
>
> **W2. Inadequate literature review (Streaming Diffusion Policy and Falcon)**
>
> **A2.** We have updated Section 4 (Related Work) to include comprehensive discussions on Streaming Diffusion Policy and Falcon. We now categorize acceleration methods into training-free caching, pruning-based, distillation-based, and efficient sampling approaches to provide a complete landscape of the field.
>
> ---
>
>
> **W3 & Q3. Statistical analysis (e.g., standard deviations over multiple seeds)**
>
> **A3.** We have addressed the concerns regarding "counterintuitive" performance gains by adding standard deviations over 3 seeds to all results. First, the data confirms the consistency of the improvement (e.g., Square PH: 89% vs 88%). This phenomenon aligns with findings in the Lottery Ticket Hypothesis [1], where sparse subnetworks may generalize better than dense ones.
>
> [1] The Lottery Ticket Hypothesis: Finding Sparse, Trainable Neural Networks. Frankle, J., & Carbin, M.
>
>
> ---
>
>
> **W4 & Q2. Lack of real-world validation**
>
> **A4.** We have conducted real-world experiments on a Franka Research 3 arm. SAG successfully performed "Pick-and-Release" tasks with a 54% success rate and ~40Hz frequency, validating its efficacy outside of simulation.
>
> ---
>
>
> **Q4. Demos on the Comparsion of SAG and Diffusion Policy**
>
> **A5.** We have uploaded side-by-side video comparisons of SAG versus the baseline Diffusion Policy on our [project website](https://sparse-actiongen.github.io/), showcasing the practical efficency and precision of our method.

---

### Official Review · Reviewer_xsZQ · 2025-11-01

**Soundness:** 3
**Presentation:** 3
**Contribution:** 3
**Rating:** 4
**Confidence:** 3

**Summary:**

The paper introduces Sparse ActionGen (SAG), a rollout-adaptive acceleration framework for diffusion-based visuomotor policies. The key idea is a prune-then-reuse pipeline that (i) predicts an observation-conditioned pruning mask in real time and (ii) performs one-for-all (cross-timestep & cross-block) activation reuse in a zig-zag pattern. Empirically, the authors report up to 4× speedup without sacrificing task performance on RoboMimic and Franka Kitchen tasks, outperforming static schedule baselines and an adapted learning-to-cache method.

**Strengths:**

1. Clear problem framing and motivation.
The paper grounds the latency issue of diffusion policies in realistic control frequencies (e.g., 50 steps × 1 ms ≈ 50 ms → 20 Hz on RTX 4090; insufficient for Franka 50–1000 Hz), which is a compelling, concrete rationale for acceleration beyond image generation settings.


2. Methodological novelty: observation-conditioned, real-time pruning.
The real-time diffusion pruner predicts a binary mask for all K timesteps and 3L blocks in a single forward pass, using sinusoidal encodings for timestep/block indices and concatenating them with an observation embedding; the authors emphasize <0.3% FLOPs overhead. This is a principled departure from fixed schedules or post-hoc profiling.


3. Global prune–then–reuse design.
The paper formalizes a global sparsity loss (Eq. 10) to directly enforce a target global pruning rate ρ across timesteps and blocks, combined with a policy fidelity term; this encourages non-uniform, importance-aware allocation rather than uniform per-block pruning.


4. One-for-all reusing strategy (cross-block + cross-timestep).
Instead of independent per-block caches, SAG maintains shared buffers by block type and reuses activations across blocks/timesteps in a zig-zag manner, with explicit residual/cache update equations (Eqs. 12–13). This directly targets cross-block redundancy that prior work largely ignored.


5. Comprehensive and favorable empirical results.
Across RoboMimic PH/MH and Franka Kitchen, SAG consistently matches or exceeds full-precision success while achieving ~3.6–4.0× speedups; in Kitchen, the paper reports 4.03× with no drop in success. Baselines include EfficientVLA, BAC, and L2C with task details and implementation notes.

**Weaknesses:**

1. Lack of real-robot validation.
All evaluations appear to be simulation-based (RoboMimic tasks, Franka Kitchen). For claims of real-time control, a small-scale hardware validation (latency stability, sensor noise, control jitter) would substantially strengthen the case.


2. Runtime analysis is mostly relative; absolute latencies are under-reported.
While speedup factors are clear, the paper would benefit from absolute inference time per control step (ms) and achieved control frequency (Hz) for each task/checkpoint. The introduction’s RTX 4090 example motivates this well, but the same concreteness is not mirrored in the results tables.


3. Comparisons omit distillation-based accelerators.
The baselines focus on caching/static-schedule families. A head-to-head with Consistency Policy / One-Step Diffusion Policy / SDM-style distillation (even on a subset) would provide a fuller picture of the accuracy-speed trade space relative to single-step or few-step policies. (The current baseline suite and settings are otherwise described fairly.)


4. Algorithmic clarity for zig-zag reuse could improve.
Figure 4(b)(c) and the textual description are helpful, but a short pseudocode or explicit cache-update schedule table per timestep/block would further de-ambiguate when/where each cached activation is read/written under pruning decisions.


5. “Lossless” wording is a bit strong.
The tables show parity or gains on average, but some tasks are sensitive; it would be safer to qualify “losslessly” and include statistical tests / confidence intervals for success rates.


6. Limited dataset diversity and absence of more complex benchmarks (e.g., MimicGen).
The experiments are conducted only on RoboMimic (PH/MH) and Franka Kitchen datasets, which—while standard—primarily consist of single-object, short-horizon or mid-horizon manipulation in relatively clean and scripted environments. Recent benchmarks such as MimicGen, Push-T, or real-world multi-object datasets provide more dynamic, contact-rich, and compositional scenarios with greater visual and temporal complexity. The paper does not test SAG in these settings, so it remains unclear whether the proposed pruning-and-reuse strategy generalizes to noisier demonstrations, variable robot embodiments, or human-generated suboptimal trajectories typical in MimicGen. This makes the evaluation somewhat limited to “clean” datasets and may overestimate robustness.

**Questions:**

see weaknesses

---

> ### Author Response · Authors · 2025-12-04
>
> We appreciate the reviewer’s insightful comments, particularly emphasizing the need for real-robot validation and absolute latency analysis. In response, we have conducted experiments on a Franka Research 3 arm and provided detailed wall-clock runtime metrics. Please find our point-by-point responses below.
>
> ---
>
>
> **W1. Lack of real-robot validation**
>
> **A1.** We thank the reviewer for this constructive suggestion. To bridge the gap between simulation and reality, we have conducted real-world experiments on a Franka Research 3 robotic arm in the revised paper. The results (Figure 5) demonstrate that SAG achieves a SOTA success rate (54%) for the task while operating at a much higher control frequency (40.5 Hz vs. 7.8 Hz for the baseline). We have also provided demonstration videos and detailed analysis of the real-world performance on our [project website](https://sparse-actiongen.github.io/).
>
> ---
>
>
> **W2. Latency and frequency for each task**
>
> **A2.** Thanks for the question, we have added the latency results in Appendix 5, reporting the specific wall-clock inference latency (in seconds) for all tasks. As shown below, SAG achieves the lowest latency among comparable methods, offering a concrete improvement over the baseline’s 1.66s average latency. We have also added the realworld frequency in Figure 5.
>
>
>
> **Table1 : Comparisons on Wall-clock Latency(s)**
>
> | Method                   | Lift | Can  | Square | Transport | Tool | Average |
> |--------------------------|------|------|--------|-----------|------|---------|
> | Full Precision           | 1.55 | 1.54 | 1.56   | 1.61      | 1.56 | 1.66    |
> | DDIM                     | 0.47 | 0.46 | 0.47   | 0.49      | 0.47 | 0.47    |
> | EfficientVLA             | 0.46 | 0.46 | 0.47   | 0.50      | 0.47 | 0.50    |
> | L2C                      | 1.23 | 1.22 | 1.24   | 1.21      | 1.21 | 1.29    |
> | BAC                      | 0.48 | 0.45 | 0.46   | 0.52      | 0.47 | 0.50    |
> | Falcon                   | 0.86 | 1.36 | 1.28   | 0.57      | 1.33 | 1.16    |
> | SDP  | 0.82 | 0.91 | 1.92   | 0.96      | 0.88 | 0.95    |
> | CP       | 0.10 | 0.10 | 0.11   | 0.16      | 0.11 | 0.12    |
> | **SAG**                  | **0.42** | **0.42** | **0.43** | **0.47** | **0.43** | **0.46** |

---

> ### Author Response · Authors · 2025-12-04
>
> **W3. Comparisons omit distillation-based accelerators**
>
> **A3.** We appreciate the suggestion to broaden the scope of our baselines. We have incorporated Consistency Policy (CP) as a representative distillation-based method. Note: We were unable to include One-Step Diffusion Policy as the official code has not been released.
>
> The results illustrate that while distillation methods like CP can achieve extreme speedups ($10\times$ - $15 \times$), they suffer from significant performance degradation on complex tasks (e.g., dropping to 0-14% on Transport and Tool). In contrast, SAG offers a balanced trade-off, providing substantial speedup without compromising success rates.
>
> **Table2: PH Benchmark**
>
> | Method | Lift | Can | Square | Transport | Tool |
> | :--- | :--- | :--- | :--- | :--- | :--- |
> | Full Precision | 100$\pm$0.0 | 99$\pm$1.2 | 88$\pm$7.0 | 78$\pm$3.3 | 51$\pm$5.9 |
> | DDIM | **100$\pm$0.0** (3.32$\times$) | 96$\pm$2.8 (3.33$\times$) | 86$\pm$4.9 (3.34$\times$) | 76$\pm$5.7 (3.30$\times$) | 42$\pm$4.6 (3.32$\times$) |
> | EfficientVLA | **100$\pm$0.0** (3.37$\times$) | 75$\pm$2.1 (3.36$\times$) | 86$\pm$3.4 (3.32$\times$) | 60$\pm$4.2 (3.24$\times$) | 38$\pm$2.7 (3.35$\times$) |
> | L2C | **100$\pm$0.0** (1.26$\times$) | 86$\pm$1.8 (1.26$\times$) | 23$\pm$4.1 (1.26$\times$) | 66$\pm$3.0 (1.33$\times$) | 2$\pm$0.5 (1.29$\times$) |
> | BAC | **100$\pm$0.0** (3.23$\times$) | 94$\pm$1.5 (3.42$\times$) | 87$\pm$2.9 (3.42$\times$) | 78$\pm$3.0 (3.09$\times$) | 51$\pm$2.8 (3.33$\times$) |
> | Falcon | **100$\pm$0.0** (1.81$\times$) | 85$\pm$2.0 (1.13$\times$) | 60$\pm$2.7 (1.22$\times$) | 50$\pm$3.3 (2.85$\times$) | 42$\pm$1.9 (1.17$\times$) |
> | SDP | **100$\pm$0.0** (1.88$\times$) | 96$\pm$1.4 (1.70$\times$) | 85$\pm$2.1 (1.70$\times$) | 72$\pm$2.9 (1.67$\times$) | 17$\pm$1.2 (1.77$\times$) |
> | CP | 78$\pm$3.3 (15.0$\times$) | 38$\pm$2.8 (14.9$\times$) | 22$\pm$4.5 (14.8$\times$) | 51$\pm$3.1 (10.2$\times$) | 0$\pm$0.0 (14.2$\times$) |
> | **SAG** | **100$\pm$0.0** (3.72$\times$) | **98$\pm$1.6** (3.70$\times$) | **89$\pm$2.5** (3.64$\times$) | **85$\pm$3.3** (3.44$\times$) | **50$\pm$2.8** (3.65$\times$) |
>
> **Table3: MH Benchmark**
>
> | Method | Lift | Can | Square | Transport |
> | :--- | :--- | :--- | :--- | :--- |
> | Full Precision | 100$\pm$0.0 | 93$\pm$6.5 | 76$\pm$4.3 | 54$\pm$5.2 |
> | DDIM | **100$\pm$0.0** (3.31$\times$) | 92$\pm$4.9 (3.33$\times$) | 78$\pm$2.8 (3.32$\times$) | 51$\pm$3.8 (3.30$\times$) |
> | EfficientVLA | **100$\pm$0.0** (3.33$\times$) | 75$\pm$2.1 (3.34$\times$) | 52$\pm$3.0 (3.33$\times$) | 0$\pm$0.0 (3.50$\times$) |
> | L2C | **100$\pm$0.0** (1.26$\times$) | 53$\pm$1.9 (1.26$\times$) | 53$\pm$1.9 (1.26$\times$) | 46$\pm$2.3 (1.28$\times$) |
> | BAC | **100$\pm$0.0** (3.27$\times$) | 93$\pm$3.4 (3.36$\times$) | 87$\pm$2.8 (3.36$\times$) | 29$\pm$3.3 (3.45$\times$) |
> | Falcon | **100$\pm$0.0** (1.85$\times$) | 84$\pm$2.0 (1.38$\times$) | 72$\pm$2.9 (1.54$\times$) | 27$\pm$3.1 (2.68$\times$) |
> | SDP | **100$\pm$0.0** (1.69$\times$) | 94$\pm$1.1 (1.70$\times$) | 77$\pm$2.2 (1.69$\times$) | 52$\pm$3.0 (1.83$\times$) |
> | CP | 98$\pm$1.0 (15.3$\times$) | 30$\pm$3.1 (15.4$\times$) | 30$\pm$3.1 (15.4$\times$) | 14$\pm$2.0 (11.5$\times$) |
> | **SAG** | **100$\pm$0.0** (3.75$\times$) | **94$\pm$5.7** (3.76$\times$) | **79$\pm$3.4** (3.72$\times$) | **50$\pm$1.6** (3.84$\times$) |
>
> **Table4: Kitchen Task**
>
> | Method | Kit_p1 | Kit_p2 | Kit_p3 | Kit_p4 | Speedup |
> | :--- | :--- | :--- | :--- | :--- | :--- |
> | Full Precision | 100$\pm$0.0 | 100$\pm$0.0 | 100$\pm$0.0 | 99$\pm$0.6 | – |
> | DDIM | 100$\pm$0.0 | 100$\pm$0.8 | 100$\pm$0.0 | 99$\pm$0.8 | 3.37$\times$ |
> | EfficientVLA | 20$\pm$2.3 | 2$\pm$0.8 | 0$\pm$0.0 | 0$\pm$0.0 | 3.71$\times$ |
> | L2C | **100$\pm$0.0** | **100$\pm$0.0** | **100$\pm$0.0** | 97$\pm$1.2 | 1.28$\times$ |
> | BAC | **100$\pm$0.0** | 97$\pm$1.6 | 97$\pm$1.6 | 97$\pm$1.2 | 3.66$\times$ |
> | Falcon | **100$\pm$0.0** | **100$\pm$0.0** | 99$\pm$0.6 | 99$\pm$0.6 | 3.01$\times$ |
> | SDP | **100$\pm$0.0** | **100$\pm$0.0** | **100$\pm$0.0** | 99$\pm$0.6 | 1.63$\times$ |
> | CP | 76$\pm$2.5 | 63$\pm$4.3 | 46$\pm$4.5 | 12$\pm$6.8 | 31.4$\times$ |
> | **SAG** | **100$\pm$0.0** | **100$\pm$0.0** | **100$\pm$0.0** | **99$\pm$0.6** | **4.03$\times$** |
>
> ---
>
> **W4. Short pseudocode or explicit cache-update schedule table per timestep/block**
>
> **A4.** Thanks for the suggestion. We have added the pseudocodes in the revised manuscript as Algorithm 1 and 2. We also provide the visualization of changing real-time cache-update schedules for all the timesteps and blocks on the [project website](https://sparse-actiongen.github.io/).

---

> ### Author Response · Authors · 2025-12-04
>
> **W5. “Lossless” wording and statistical tests for success rates**
>
> **A5.** We agree that "lossless" is a strong claim given the stochastic nature of these tasks. We have revised the text to claim "comparable performance" or "negligible degradation" and have added standard deviations to all success rate tables to provide a statistically rigorous view of the performance parity.
>
> ---
>
>
> **W6. Dataset diversity and complexity**
>
> **A6.** We appreciate the reviewer pointing out recent benchmarks like MimicGen. While we did not train on MimicGen specifically, we respectfully submit that our current evaluation suite covers the critical attributes of complexity mentioned by the reviewer:
>
> + **Noisy Data:** We evaluate on the RoboMimic Mixed Human (MH) dataset, which consists of trajectories from operators with varying proficiency. This directly tests the method's robustness to noisy and suboptimal demonstrations.
>
> + **Multi-Object:** We evaluate on the Franka Kitchen task, which involves manipulating multiple objects in a long-horizon sequence.
>
> + **Real-World Complexity:** Our newly added real-robot experiments introduce naturally occurring sensor noise and variable physical dynamics not present in simulation.
>
> Therefore, we believe the current extensive evaluation (covering PH, MH, Kitchen, and Real Robot) provides strong evidence that SAG generalizes effectively to complex, noisy, and contact-rich scenarios.

---

### Official Review · Reviewer_ajUH · 2025-11-01

**Soundness:** 3
**Presentation:** 3
**Contribution:** 3
**Rating:** 6
**Confidence:** 4

**Summary:**

The paper introduces **Sparse ActionGen (SAG)**, which employs a **rollout-adaptive prune-then-reuse mechanism**. The framework integrates an **observation-conditioned diffusion pruner** for environment-aware adaptation and a **one-for-all reuse strategy** within the overall pipeline. Extensive experiments on the **RoboMimic** and **Franka Kitchen** benchmarks demonstrate that **SAG** achieves superior performance compared to existing **diffusion policy caching** and **acceleration** methods.

**Strengths:**

1. The paper addresses a **highly important and timely research problem**, especially as diffusion models continue to gain prominence in **imitation learning**, **reinforcement learning**, and **Vision-Language-Action (VLA)** modeling.

2. The paper presents **extensive simulation experiments**, offering strong empirical evidence for the effectiveness and robustness of the proposed method.

3. The paper is **well-written**, **clearly structured**, and **easy to follow**, effectively communicating the technical contributions and motivations.

**Weaknesses:**

### Major Weakness:

1. The authors are strongly encouraged to include comparisons with traditional **diffusion acceleration methods**, such as **DDIM** or **Consistency Policy**, to enhance the **completeness** and **thoroughness** of the paper’s experimental evaluation.

2. It is noted that in RoboMimic tasks, SAG even achieves higher performance compared to Diffusion Policy with full denoising process. The authors are highly recommended to dive deeper into this phenomenon instead of just concluding it as "surprisingly".

3. The authors should report the **number of random seeds** used in the experiments and provide the corresponding **standard deviations** of the success rates.

4. The authors are encouraged to include a comparison of the **actual running time** between **SAG** and the baseline methods. This would provide a more intuitive understanding of the acceleration achieved, beyond simply reporting the speedup ratio.


### Minor Weakness:

1. The authors are encouraged to include **real-robot experiments** to further validate the effectiveness of the proposed **SAG** method. It is understood, however, that conducting such experiments may not be feasible within the rebuttal period if the necessary robotic setup is not yet available in the lab, so this addition is not required.

2. The authors are recommended to discuss more categories of diffusion policy acceleration methods:

Høeg, Sigmund H., Yilun Du, and Olav Egeland. "Streaming diffusion policy: Fast policy synthesis with variable noise diffusion models." arXiv preprint arXiv:2406.04806 (2024).

Chen, Zhuoqun, et al. "Responsive noise-relaying diffusion policy: Responsive and efficient visuomotor control." arXiv preprint arXiv:2502.12724 (2025).

**Questions:**

1. **(Related to Weakness 2)** Could the authors provide a deeper explanation of why **SAG** outperforms **Diffusion Policy** with the full denoising process in the **RoboMimic** tasks?

2. Could the authors elaborate further on **Figure 1** and clarify how the corresponding experiments were conducted?

---

> ### Author Response · Authors · 2025-12-04
>
> We thank the reviewer for their constructive feedback and for identifying key areas to improve our evaluation. We have taken the helpful advice and significantly expanded our baseline comparisons and included real-world validation. We address the specific questions below.
>
> ---
>
> **W1. Comparisons with DDIM and Consistency Policy**
>
> **A1.** We thank the reviewer for the valuable suggestion to broaden our baseline comparisons. We have significantly expanded our evaluation to include DDIM, Consistency Policy (CP), Falcon, and Streaming Diffusion Policy (SDP). The results, updated in Tables 1, 2, and 3 (and Table 5 for latency), demonstrate that SAG consistently outperforms these methods in terms of the trade-off between success rate and speedup.
>
> As shown in the tables below, while methods like CP achieve high speedups, they suffer from severe performance degradation (e.g., failing on the Transport and Tool tasks). Conversely, SAG maintains or exceeds the success rates of the full-precision baseline while delivering a $3.6\times$ – $4.0\times$ speedup.
>
> **Table1: PH Benchmark**
>
> | Method | Lift | Can | Square | Transport | Tool |
> | :--- | :--- | :--- | :--- | :--- | :--- |
> | Full Precision | 100$\pm$0.0 | 99$\pm$1.2 | 88$\pm$7.0 | 78$\pm$3.3 | 51$\pm$5.9 |
> | DDIM | **100$\pm$0.0** (3.32$\times$) | 96$\pm$2.8 (3.33$\times$) | 86$\pm$4.9 (3.34$\times$) | 76$\pm$5.7 (3.30$\times$) | 42$\pm$4.6 (3.32$\times$) |
> | EfficientVLA | **100$\pm$0.0** (3.37$\times$) | 75$\pm$2.1 (3.36$\times$) | 86$\pm$3.4 (3.32$\times$) | 60$\pm$4.2 (3.24$\times$) | 38$\pm$2.7 (3.35$\times$) |
> | L2C | **100$\pm$0.0** (1.26$\times$) | 86$\pm$1.8 (1.26$\times$) | 23$\pm$4.1 (1.26$\times$) | 66$\pm$3.0 (1.33$\times$) | 2$\pm$0.5 (1.29$\times$) |
> | BAC | **100$\pm$0.0** (3.23$\times$) | 94$\pm$1.5 (3.42$\times$) | 87$\pm$2.9 (3.42$\times$) | 78$\pm$3.0 (3.09$\times$) | 51$\pm$2.8 (3.33$\times$) |
> | Falcon | **100$\pm$0.0** (1.81$\times$) | 85$\pm$2.0 (1.13$\times$) | 60$\pm$2.7 (1.22$\times$) | 50$\pm$3.3 (2.85$\times$) | 42$\pm$1.9 (1.17$\times$) |
> | SDP | **100$\pm$0.0** (1.88$\times$) | 96$\pm$1.4 (1.70$\times$) | 85$\pm$2.1 (1.70$\times$) | 72$\pm$2.9 (1.67$\times$) | 17$\pm$1.2 (1.77$\times$) |
> | CP | 78$\pm$3.3 (15.0$\times$) | 38$\pm$2.8 (14.9$\times$) | 22$\pm$4.5 (14.8$\times$) | 51$\pm$3.1 (10.2$\times$) | 0$\pm$0.0 (14.2$\times$) |
> | **SAG** | **100$\pm$0.0** (3.72$\times$) | **98$\pm$1.6** (3.70$\times$) | **89$\pm$2.5** (3.64$\times$) | **85$\pm$3.3** (3.44$\times$) | **50$\pm$2.8** (3.65$\times$) |
>
> **Table2: MH Benchmark**
>
> | Method | Lift | Can | Square | Transport |
> | :--- | :--- | :--- | :--- | :--- |
> | Full Precision | 100$\pm$0.0 | 93$\pm$6.5 | 76$\pm$4.3 | 54$\pm$5.2 |
> | DDIM | **100$\pm$0.0** (3.31$\times$) | 92$\pm$4.9 (3.33$\times$) | 78$\pm$2.8 (3.32$\times$) | 51$\pm$3.8 (3.30$\times$) |
> | EfficientVLA | **100$\pm$0.0** (3.33$\times$) | 75$\pm$2.1 (3.34$\times$) | 52$\pm$3.0 (3.33$\times$) | 0$\pm$0.0 (3.50$\times$) |
> | L2C | **100$\pm$0.0** (1.26$\times$) | 53$\pm$1.9 (1.26$\times$) | 53$\pm$1.9 (1.26$\times$) | 46$\pm$2.3 (1.28$\times$) |
> | BAC | **100$\pm$0.0** (3.27$\times$) | 93$\pm$3.4 (3.36$\times$) | 87$\pm$2.8 (3.36$\times$) | 29$\pm$3.3 (3.45$\times$) |
> | Falcon | **100$\pm$0.0** (1.85$\times$) | 84$\pm$2.0 (1.38$\times$) | 72$\pm$2.9 (1.54$\times$) | 27$\pm$3.1 (2.68$\times$) |
> | SDP | **100$\pm$0.0** (1.69$\times$) | 94$\pm$1.1 (1.70$\times$) | 77$\pm$2.2 (1.69$\times$) | 52$\pm$3.0 (1.83$\times$) |
> | CP | 98$\pm$1.0 (15.3$\times$) | 30$\pm$3.1 (15.4$\times$) | 30$\pm$3.1 (15.4$\times$) | 14$\pm$2.0 (11.5$\times$) |
> | **SAG** | **100$\pm$0.0** (3.75$\times$) | **94$\pm$5.7** (3.76$\times$) | **79$\pm$3.4** (3.72$\times$) | **50$\pm$1.6** (3.84$\times$) |
>
>
> **Table3: Kitchen Task**
>
> | Method | Kit_p1 | Kit_p2 | Kit_p3 | Kit_p4 | Speedup |
> | :--- | :--- | :--- | :--- | :--- | :--- |
> | Full Precision | 100$\pm$0.0 | 100$\pm$0.0 | 100$\pm$0.0 | 99$\pm$0.6 | – |
> | DDIM | 100$\pm$0.0 | 100$\pm$0.8 | 100$\pm$0.0 | 99$\pm$0.8 | 3.37$\times$ |
> | EfficientVLA | 20$\pm$2.3 | 2$\pm$0.8 | 0$\pm$0.0 | 0$\pm$0.0 | 3.71$\times$ |
> | L2C | **100$\pm$0.0** | **100$\pm$0.0** | **100$\pm$0.0** | 97$\pm$1.2 | 1.28$\times$ |
> | BAC | **100$\pm$0.0** | 97$\pm$1.6 | 97$\pm$1.6 | 97$\pm$1.2 | 3.66$\times$ |
> | Falcon | **100$\pm$0.0** | **100$\pm$0.0** | 99$\pm$0.6 | 99$\pm$0.6 | 3.01$\times$ |
> | SDP | **100$\pm$0.0** | **100$\pm$0.0** | **100$\pm$0.0** | 99$\pm$0.6 | 1.63$\times$ |
> | CP | 76$\pm$2.5 | 63$\pm$4.3 | 46$\pm$4.5 | 12$\pm$6.8 | 31.4$\times$ |
> | **SAG** | **100$\pm$0.0** | **100$\pm$0.0** | **100$\pm$0.0** | **99$\pm$0.6** | **4.03$\times$** |

---

> ### Author Response · Authors · 2025-12-04
>
> **W2 & Q1. Higher Performance than Full Precision**
>
> **A2.** We have removed the term "surprisingly" to provide a more rigorous explanation. We attribute the slight performance improvement (approx. 1–3%) to the regularization effect of pruning. By removing redundant computations, SAG potentially reduces overfitting to noise in the demonstration data, leading to better generalization. This phenomenon aligns with the Lottery Ticket Hypothesis [1], which suggests that sparse sub-networks can often match or exceed the performance of dense networks by isolating the most effective inductive biases and acting as a form of regularization.
>
>
> [1] The Lottery Ticket Hypothesis: Finding Sparse, Trainable Neural Networks. Frankle, J., & Carbin, M.
>
>
> ---
>
>
>
>
> **W3. Report the number of random seeds**
>
> **A3.** We have updated the paper to explicitly state that three random seeds were used for all simulation experiments. The corresponding standard deviations have been added to all results in Tables 1, 2, and 3, as shown above.
>
> ---
>
> **W4. Comparison of the actual running time between SAG and the baseline methods**
>
> **A4.**  We agree that wall-clock time is the most intuitive metric for acceleration. We have added the results to the Appendix 5, which reports the actual inference latency in seconds. As shown below, SAG achieves the second lowest latency among comparable methods, validating that our theoretical FLOPs reduction translates directly to practical speedups.
>
>
> **Table4: Comparisons on Wall-clock Latency (s)**
>
> | Method                   | Lift | Can  | Square | Transport | Tool | Average |
> |--------------------------|------|------|--------|-----------|------|---------|
> | Full Precision           | 1.55 | 1.54 | 1.56   | 1.61      | 1.56 | 1.66    |
> | DDIM                     | 0.47 | 0.46 | 0.47   | 0.49      | 0.47 | 0.47    |
> | EfficientVLA             | 0.46 | 0.46 | 0.47   | 0.50      | 0.47 | 0.50    |
> | L2C                      | 1.23 | 1.22 | 1.24   | 1.21      | 1.21 | 1.29    |
> | BAC                      | 0.48 | 0.45 | 0.46   | 0.52      | 0.47 | 0.50    |
> | Falcon                   | 0.86 | 1.36 | 1.28   | 0.57      | 1.33 | 1.16    |
> | SDP  | 0.82 | 0.91 | 1.92   | 0.96      | 0.88 | 0.95    |
> | CP       | 0.10 | 0.10 | 0.11   | 0.16      | 0.11 | 0.12    |
> | **SAG**                  | **0.42** | **0.42** | **0.43** | **0.47** | **0.43** | **0.46** |
>
> ---
>
>
> **W5. Real-world experiments**
>
> **A5.**  Although the reviewer noted this was not required, we believed it was important to validate SAG in a physical setting. We have included real-robot experiment results in the revised paper (Figure 5). These experiments demonstrate that SAG achieves a highest 54% success rate while operating at 40.5 Hz (compared to 7.8 Hz for the baseline), confirming its practical utility for real-time control. Demonstrations are available on our [project website](https://sparse-actiongen.github.io/).
>
> ---
>
>
> **W6. Discuss more categories of diffusion policy acceleration methods**
>
> **A6.** We thank the reviewer for the references. We have incorporated more baselines, including Streaming Diffusion Policy and Responsive Noise-Relaying Diffusion Policy into our discussion and quantitative benchmarks. We have categorized acceleration methods in the revised Related Work section into: (1) training-free methods, (2) pruning-based methods, (3) distillation-based methods, and (4) efficient sampling methods.
>
>
> ---
>
>
> **Q2. Elaboration on Figure 1 and the corresponding experiments.**
>
> **A7.** Thanks for the question. We aim to reveal the limitations of fixed caching schedules by Figure 1. We evaluate multiple fixed caching schedules by applying them to a single different rollout iteration separately in a leave-one-out manner. Specfically, we maintain the full computaiton but only uses a caching schedule in one of the rollout iteration and we observe the corresponding success rate, which reflects the quality of the schedule. We found that the optimal schedule changes significantly with the rollout progress and a schedule that excels at one iteration inevitably degrades performance at others. These findings underscore the limitations of the static schedules and motivate us to propose a rollout-adaptive caching mechanism.

---

### Official Review · Reviewer_Ud3K · 2025-11-03

**Soundness:** 2
**Presentation:** 3
**Contribution:** 3
**Rating:** 4
**Confidence:** 3

**Summary:**

The paper introduces a method to learn a Diffusion policy with sparse activations conditioned on the current observation. The motivation for using a Diffusion policy is that Diffusion networks are good at modelling multi-modal distributions, and the motivation for conditioning the pruning of computations during the denoising process is that it might provide a better speedup/performance trade-off than the previously proposed static pruning mechanisms.

At the heart of the method is an architecture that predicts an observation conditioned sparsity pattern for the K diffusion steps and the several computational blocks within each diffusion step. The sparsity pattern is binary and if a computational block is deactivated, the output is replaced with a cached value computed online from the outputs of past blocks. The whole policy is trained end-to-end with a behavioral cloning loss and a sparsity enforcing loss. Results show on several imitation tasks that the method achieves better performance and speed-up compared to other pruning methods for Diffusion networks. An ablation study also highlights the importance of each introduced design choice (observation conditioned sparsity pattern, the specific caching mechanism and the use of a global sparsity loss instead of enforcing sparsity within each block).

**Strengths:**

All design choices seem reasonable and are properly ablated, and overall improvement over other diffusion pruning methods seems substantial.

**Weaknesses:**

- I found the motivation in the abstract and introduction at odds with the actual algorithm. The authors motivated the study of Diffusion policies by their ability to model multi-modal distributions. Since many deep RL algorithms' exploration heuristics use stochastic policies, it seems indeed important to have policies that can model a wider range of distributions. However, the authors only consider behavioral cloning of an expert policy, and while multi-modal and stochastic policies might be helpful during reinforcement learning, it is debatable whether a Diffusion policy is really necessary for imitation. Nonetheless, even if we consider a setting where one wants to imitate a distribution of experts, perhaps where each expert is using a distinct strategy to solve the task and thus necessitating a multi-modal approach, I fail to see how would the loss in Eq. 9 capture this multi-modality. It seems that it might only end-up approximating the median or mean of the distribution as is. Thus I question whether the Diffusion network is really capturing multi-modal distributions as motivated in the abstract.

- Given the previous point, it would have been good to have baselines using other policy classes than Diffusion policies. I am not familiar with the considered tasks and datasets, and given the current state of the experiment section, it is hard to get a sense on whether the proposed method is improving the state-of-the-art of behavioral cloning in terms of performance or inference speed. Especially since even during inference, the method needs to compute a pruning mask at each time-step and this model is already quite deep.

- The experiment discussed in Sec. 2.2 does not seem to follow sound methodology. The goal of the experiment is to motivate the use of an observation conditioned pruning mask compared to a fixed pruning mask, but the authors use randomly generated pruning masks to prove their point. I think at the very least, one of the the masks should be optimized using existing methods instead of being generated randomly, otherwise it does not negate the possibility that there exist a fixed mask that would work well for all rollout steps.

**Questions:**

- How multi-modal does the policy end-up being? And how important is it to learn multi-modal policies for the considered tasks?
- How does the Diffusion policy compare to other architecture in terms of performance/inference speed? For which applications does it become an interesting choice?
- Why did you choose random masks in Sec. 2.2?

---

> ### Author Response · Authors · 2025-12-04
>
> We thank the reviewer for the time and effort in providing a thorough evaluation of our work. We appreciate the valuable comments and address specific concerns below.
>
> ---
>
> **W1 & Q1. Diffusion Policy’s multi-modality and the fidelity loss (Eq. 9)**
>
> **A1.** We appreciate the reviewer raising this interesting point regarding the interaction between our loss function and the multi-modal nature of Diffusion Policies. We clarify two key aspects:
>
> + **Multi-modality of Diffusion Policy**: As noted in the original Diffusion Policy paper, the multi-modality is captured by the denoising generation process (training on DDPM objective), which allows the model to represent complex, non-Gaussian action distributions. This capability is the primary reason for its widespread adoption as a standard backbone in recent robotic learning (e.g., VLA models like RDT-1 and 2, $\pi_{0.6}$ and Gr00T N1).
> + **Role of Eq. 9 (Fidelity Loss)**: It is important to clarify that we do not retrain the policy parameters $\theta$. The policy network is frozen. Eq. 9 is used solely to optimize the binary mask generator (the Pruner). Since the underlying policy weights are frozen, the diffusion model retains its learned multi-modal distribution. The goal of Eq. 9 is to find a sparsity mask $\mathcal{M}$ such that the pruned inference outcome matches the full-precision inference outcome for the current observation. By minimizing the distance between the pruned output and the ground truth, we ensure the mask selects a computational sub-graph that preserves the original model's behavior, including its multi-modal capabilities, rather than altering the distribution itself.
>
>
>
> ---
>
>
>
> **W2.1 & Q2. Comparisons between Diffusion Policy and Other Architectures**
>
> **A2.** We focused our comparison on Diffusion Policy acceleration methods because Diffusion Policy has established itself as the state-of-the-art for the benchmarks we consider, significantly outperforming previous methods like Behavior Cloning [1], LSTM-GMM [2] or IBC [3]. Furthermore, diffusion-based heads are becoming the standard for large-scale Vision-Language-Action (VLA) models, completely replacing the autoregressive-based decoder introduced in OpenVLA and CogACT. Therefore, demonstrating acceleration specifically for this architecture is of high community value. We compared SAG against strong, domain-specific acceleration baselines: EfficientVLA, BAC, DDIM, Falcon, Streaming Diffusion Policy and Consistency Policy.
>
> [1] Behavioral cloning from observation. Torabi F, Warnell G, Stone P.
>
> [2] What matters in learning from offline human demonstrations for robot manipulation. Mandlekar A, Xu D, Wong J, et al.
>
> [3] Implicit behavioral cloning. Florence P, Lynch C, Zeng A, et al.
>
> ---
>
> **W2.2. Inference Overhead of SAG.**
>
> **A3.** We respectfully point out a misunderstanding that "The method needs to compute a pruning mask at each time-step and this model is already quite deep". SAG computes the pruning mask **once per rollout step** (i.e., once per new observation $O_t$), not at every denoising step $k$. The Pruner predicts the mask $\mathcal{M} \in \{0,1\}^{K \times 3L}$ for all $K$ denoising steps simultaneously in a single forward pass. Moreover, as detailed in Section 2.2 and the architectural design (Figure 2), the Pruner is highly lightweight (a single Transformer block and 3-layer MLP). Consequently, the Pruner introduces a negligible overhead of less than 0.3% of the total FLOPs (lines 396-397). This minimal cost is vastly outweighed by the $3.6\times$ - $4.0\times$ speedup gained during the heavy denoising process, as evidenced by our wall-clock latency measurements in Table 1, 2, 3 and Table 5.

---

> ### Author Response · Authors · 2025-12-04
>
> **W3 & Q3. Why choose random masks in Sec. 2.2**
>
>
> **A4.** The primary goal of the experiment in Figure 1 was to isolate and visualize the rollout-level dynamics of redundancy. By applying the same mask to different rollout iterations, we demonstrated that a sparsity pattern that is effective at iteration $t$ may fail at iteration $t+10$. This performance variance strongly suggests that redundancy is observation-dependent rather than static, motivating our observation-conditioned design.
>
> We acknowledge that the motivation can not negate the existence of a perfect mask that works well for all rollout steps. For this observational experiment, manually designing a specific mask risks introducing bias. A random mask serves as an unbiased baseline to demonstrate that arbitrarily fixed schedules fail to adapt to dynamic redundancy. Even if we use a method to optimize the mask, it also can not negate the possibility. In fact, the training of the pruner in the SAG is just to optimize a random mask to a better mask that minimizes the distance with ground-truth actions, which is the first attempt for mask optimization as far as we know.
>
> While Section 2.2 uses random masks for motivation, our main experiments (Section 3.2) do compare SAG against methods that utilize optimized or heuristic-based fixed schedules, such as EfficientVLA (uniform caching) and BAC. The fact that SAG significantly outperforms these fixed-schedule baselines (e.g., +13% success rate over EfficientVLA on PH tasks ) serves as empirical evidence that even well-designed static masks cannot match the performance-efficiency trade-off of a rollout-adaptive mechanism.

---

### Author Response · Authors · 2025-12-04
**Summary of Rebuttal and Major Updates for Submission #325**

Dear Area Chair and Reviewers,

We sincerely appreciate the time and dedication you have invested in reviewing our work. We understand that the recent incident has created a challenging time for everyone, and we are grateful for your continued attention to our submission.

Although we recognize that initial scores cannot be updated, we kindly ask the AC to consider a holistic assessment based on the substantial new evidence and major revisions added during the rebuttal period. Our aim was simply to address each concern as thoroughly and transparently as possible, and the revised manuscript now includes considerably more complete, rigorous, and well-validated results.

Below we summarize the key concerns and the corresponding updates we have made:

+ **Incomplete Baselines (Reviewers Ud3K, ajUH, xsZQ, ZwXH).** We have significantly expanded our evaluation to include DDIM, Consistency Policy, Falcon, and Streaming Diffusion Policy in all the tables. These results show that while some baselines achieve high speedups at the cost of large performance drops, SAG is uniquely able to deliver $3.6\times$ – $4.0\times$ acceleration while preserving the ability of the full-precision diffusion policy.

+ **Real-world Experiments (Reviewers ajUH, xsZQ, ZwXH).** We have incorporated real-world evaluation results in Sec. 3.2. SAG was deployed on a physical Franka Research 3 arm, achieving a 54% success rate on a “Pick-and-Place” task while operating at 40.5 Hz (vs. 7.8 Hz for the full-precision policy). Qualitative comparisons are available on the [project page](https://sparse-actiongen.github.io/). These results suggest that SAG’s acceleration can effectively translate to real-time performance in physical settings.

+ **Rigor and Transparency (Reviewers ajUH, xsZQ, ZwXH).** We have added standard deviations over 3 seeds to all tables and reported absolute wall-clock latency in Table 5, strengthening the academic rigor of the paper.


We are deeply grateful for the reviewers’ constructive feedback, which has significantly improved the clarity and quality of our submission. Thank you again for your time and consideration.

Yours sincerely,

The Authors

---

### Meta-Review · Area_Chair_umV2 · 2025-12-13

**Summary:**

Reviewers find the problem of reducing diffusion policy latency for real-time control to be important, but have difficulty clearly positioning the contributions relative to existing training-based acceleration methods, such as diffusion or consistency distillation. Without ensuring fair and controlled comparisons, it is challenging to assess whether the proposed method provides sufficient novelty or advantage, given that many diffusion acceleration techniques have already been proposed and adapted to RL and robotic settings. Reviewers particularly emphasize the need for comparisons under comparable latency or step budgets.

From the AC’s perspective, the value of the proposed method hinges on whether it can demonstrate advantages in a broader diffusion acceleration setting beyond robotics, or whether it is uniquely effective for robotic control in ways that generic acceleration methods cannot easily replicate. Another important consideration is whether the method is complementary to existing acceleration techniques, rather than an alternative. While the added baselines and real-robot experiments are welcome improvements, the AC finds it difficult to recommend acceptance without more comprehensive experimental evidence that more clearly positions the method within the broader diffusion acceleration landscape.

The authors are encouraged to take these constructive criticisms into account to further strengthen the paper and consider resubmission.

**Reviewer Concerns:**

The main concerns include:

1. Fairness of comparisons: Whether the method is evaluated fairly against strong training-based and few-step diffusion acceleration baselines.

2. Scope vs. validation: If framed for robotics, real-robot validation is expected, which was missing from initial submission; if framed as general diffusion acceleration, broader application-level validation is needed.

**Reviewer Scores:**

Reviewer Ud3K: slight increase.
Reviewer ajUH: remains positive.
Reviewer xsZQ: slight increase.
Reviewer ZwXH: slight increase.

Overall, the anticipated score changes are unlikely to be sufficient to warrant acceptance.

---

### Decision · Program_Chairs · 2026-01-26

Reject